# Auditing Black-Box LLM APIs with a Rank-Based Uniformity Test

**Xiaoyuan Zhu**[1]   **Yaowen Ye**[2†]   **Tianyi Qiu**[3†]   **Hanlin Zhu**[2‡]   **Sijun Tan**[2‡]
**Ajraf Mannan**[1]   **Jonathan Michala**[1]   **Raluca Ada Popa**[2]   **Willie Neiswanger**[1]

[1]University of Southern California [2]University of California, Berkeley [3]Peking University

xzhu9839@usc.edu, elwin@berkeley.edu, qiutianyi.qty@gmail.com
{hanlinzhu, sijuntan}@berkeley.edu, {amannan, michala}@usc.edu
raluca@eecs.berkeley.edu, neiswang@usc.edu

† Co-second authors. ‡ Co-third authors.

## ABSTRACT

As API access becomes a primary interface to large language models (LLMs), users often interact with black-box systems that offer little transparency into the deployed model. To reduce costs or maliciously alter model behaviors, API providers may discreetly serve quantized or fine-tuned variants, which can degrade performance and compromise safety. Detecting such substitutions is difficult, as users lack access to model weights and, in most cases, even output logits. To tackle this problem, we propose a Rank-based Uniformity Test (RUT) that can verify the behavioral equality of a black-box LLM to a locally deployed authentic model. Our method is accurate, query-efficient, and avoids detectable query patterns, making it robust to adversarial providers that reroute or mix responses upon the detection of testing attempts. We evaluate the approach across diverse query domains and threat scenarios, including quantization, harmful fine-tuning, jailbreak prompts, and full model substitution, showing that it consistently achieves superior detection power over prior methods under constrained query budgets. We release our code at https://github.com/xyzhu123/RUT.

## 1 INTRODUCTION

APIs have become a central access point for large language models (LLMs) in consumer applications, enterprise tools, and research workflows (Anysphere Inc., 2025; Yun et al., 2025; ResearchFlow, 2025). However, while users can query black-box APIs, they have little to no visibility into the underlying model implementation. Combined with the high cost of serving large models and the latency pressure to reduce time-to-first-token (TTFT), API providers are incentivized to deploy smaller or quantized variants of the original model to cut costs. Such modifications, while opaque to end users, can degrade model performance and introduce safety risks (Egashira et al., 2024). In more concerning cases, providers may incorporate harmful fine-tuning, jailbreak-enabling system prompts, or even misconfigured system components without realizing it (mirpo, 2025).

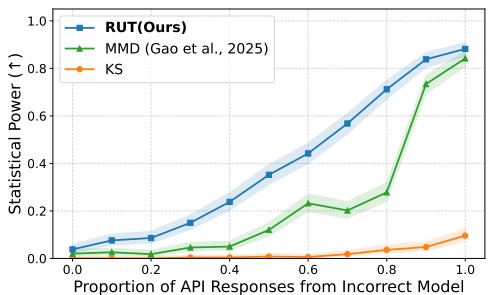

Figure 1: Statistical power of different methods in detecting substitution of the Gemma-2-9b-it with its 4-bit quantized variant, as the proportion of API responses from the quantized model increases. Our method significantly outperforms MMD (Gao et al., 2025) and the Kolmogorov–Smirnov (KS) baseline.

These risks highlight the need for *LLM API auditing*—the task of checking whether a deployed model is as claimed. Yet this is particularly challenging in the black-box setting: users typically lack access to model weights and receive only limited metadata (e.g., top-5 token log-probabilities). This necessitates detection methods that rely solely on observed outputs. However, even such output-

level methods face potential evasion: if the detection relies on invoking an LLM API with specially constructed query distributions, a dishonest API provider could detect the special pattern and reroute those queries to the original model they claim to serve. Worse still, even without knowing the detection strategy, an API provider could mix multiple models, making the response distribution harder to distinguish.

In this work, we focus on auditing API providers that serve **open-weight models**, and formulate the LLM API auditing as a *model equality test* following Gao et al. (2025): given query access to a target API and a certified reference model of the expected configurations, the goal is to determine if the two produce statistically indistinguishable outputs on shared prompts. The auditor is assumed to have full access to a faithful reference implementation of the claimed model, such as its decoding parameters and logit outputs.

We propose that a model equality test for auditing LLM APIs must satisfy three key criteria: *accuracy*, *query efficiency*, and *robustness* to adversarial attacks. Accuracy reflects how reliably a test can be used in practice. Query efficiency is critical for reducing operational overhead, which incentivizes more audits to ensure API models' integrity. Robustness is equally essential for real-world deployment, where audits must both evade detection by adversarial API providers and remain effective under targeted attacks.

While several methods have been proposed for model equality testing, they each fall short in one or more of these criteria (Table 1). Existing methods include Maximum Mean Discrepancy (MMD) (Gao et al., 2025), trained text classifiers (Sun et al., 2025), identity prompting (Huang et al., 2025), and benchmark performance comparison (Chen et al., 2023a). However, Sun et al. (2025) require prohibitively many API queries; Huang et al. (2025) fail to capture model variations such as size, version, or quantization (Cai et al., 2025); and Gao et al. (2025) and Chen et al. (2023a) rely on special query distributions that can be adversarially detected and bypassed via prompt caching (Gu et al., 2025).

Driven by these limitations, we propose a Rank-based Uniformity Test (RUT)—an asymmetric two-sample hypothesis test that addresses all three criteria simultaneously. In RUT, we sample one response from the target API and multiple responses from the reference model for each prompt, then compute the rank percentile of the API output within the reference distribution. If the target and reference models are identical, the percentiles should follow a uniform distribution. We detect deviations using the Cramér–von Mises test (Cramér, 1928). Our method requires only a single API call per prompt, operates effectively on real-world, user-like queries, and avoids detectable patterns that adversarial providers might exploit.

Table 1: Comparison of LLM auditing methods by *accuracy* (Acc.), *query-efficiency* (Q-Eff.), and *robustness to adversarial providers* (Rob.).

| Method | Acc. | Q-Eff. | Rob. |
|---|---|---|---|
| **RUT (Ours)** | ✓ | ✓ | ✓ |
| MMD (Gao et al., 2025) | ✓ | ✓ | ✗ |
| Classifier (Sun et al., 2025) | ✗ | ✗ | ✓ |
| Identity-prompting (Huang et al., 2025) | ✗ | ✓ | ✗ |
| Benchmark (Chen et al., 2023a) | ✗ | ✗ | ✗ |

We evaluate RUT across a range of adversarial scenarios in which the API provider secretly substitutes the claimed model with an alternative. In Section 5.2, we study the case where the substitute is a quantized version of the original model. In Section 5.3, we test models augmented with a hidden jailbreaking system prompt. In Section 5.4, we examine models finetuned on instruction-following data. In Section 5.5, we consider substitution with a completely different model. Finally, in Section 5.6, we test the methods on additional query distributions of math and coding.

Under a fixed API query budget, we find RUT outperforms both MMD and a Kolmogorov–Smirnov test (KS) baseline across all settings. It consistently achieves higher statistical power and shows greater robustness to probabilistic substitution attacks (Figure 1). Moreover, when applied to five real-world API-deployed models (Section 5.7), our method yields detection results closely aligned with other methods and is more robust over string-based metrics on minor decoding mismatches.

To summarize, the main contributions of our work include:

1. **A novel test for auditing LLM APIs.** We propose RUT, an asymmetric two-sample-test that needs only one API call per prompt and operates effectively on natural queries, achieving query efficiency and by-design robustness to adversarial providers.

2. **Empirical validation across diverse threat models.** We validate RUT under diverse settings, including quantization, jailbreaking, SFT, and full model replacement.

3. **Cross-validated audit of live commercial endpoints.** We benchmark RUT side-by-side with established tests (MMD and KS) on three major public LLM APIs and demonstrate its practicality in real-world black-box settings.

## 2 RELATED WORK

**LLM fingerprinting.** Fingerprinting approaches focus on identifying LLMs by analyzing their outputs. Active fingerprinting involves injecting backdoor-like behavior (Xu et al., 2024) into an LLM via finetuning, embedding watermarks (Kirchenbauer et al., 2023; Ren et al., 2023) into a model's text generation process, or intentionally crafting prompts to elicit unique outputs from different LLMs (Pasquini et al., 2024). Passive fingerprinting, on the other hand, focuses on analyzing the inherent patterns in LLM-generated text (Su et al., 2023; Fu et al., 2025; Alhazbi et al., 2025). This builds on the observation that LLMs expose rich "idiosyncrasies"—distributional quirks that allow classifiers to identify a model (Sun et al., 2025). While related, fingerprinting aims to authenticate the origin of the model and prevent publisher overclaim. Consequently, fingerprints are designed to remain stable under fine-tuning or deployment changes. This is opposite to our auditing objective. Prior work (Cai et al., 2025) also shows that fingerprinting methods fail to detect quantized substitutions.

**Auditing LLM APIs.** A growing body of work investigates whether black-box APIs faithfully serve the advertised model. The most straightforward audit is to evaluate models' benchmark performance (art, 2025; Eyuboglu et al., 2024; Chen et al., 2023b), but raw performance alone cannot expose covert substitutions or partial routing. Gao et al. (2025) formalizes the problem as *Model Equality Testing* and shows that a kernel-MMD test can already flag public endpoints that deviate from their open-weight checkpoints. Concurrently to our work, Cai et al. (2025) investigate the model substitution setting and show that API providers can evade detection through strategies such as model quantization, randomized substitution, and benchmark evasion. Building on these insights, we propose a method that is more robust to such attacks and extend the threat model to include a broader range of realistic scenarios, such as jailbroken or maliciously finetuned models.

## 3 PROBLEM FORMULATION

In this section, we formalize the LLM API auditing problem as a black-box model equality test between a target API and a fully accessible reference model.

**Models.** We denote an LLM as a conditional distribution $\pi(y|x; \varphi)$ over text output $y \in \mathcal{Y}$ given an input prompt $x \in \mathcal{X}$ and decoding parameters $\varphi$ (e.g., temperature, top-$p$). In all experiments, we control the decoding parameters to be the same across models. Two such models are involved in the test. The *reference model* $\pi_{\text{ref}}$ is fully accessible locally; the verifier can draw an unlimited number of samples $y_{\text{ref}} \sim \pi_{\text{ref}}(\cdot \mid x; \varphi)$ for any $x$ and obtain relevant logits. The *target model* $\pi_{\text{tgt}}$ is exposed only through a remote API that returns exactly one completion $y_{\text{tgt}} \sim \pi_{\text{tgt}}(\cdot \mid x; \varphi)$ per call and charges for every invocation. The total API budget is $N$ prompts.

**Hypotheses.** The model equality test is cast as a composite two-sample hypothesis test. The null hypothesis

$$H_0: \quad \pi_{\text{tgt}}(\cdot \mid x; \varphi) = \pi_{\text{ref}}(\cdot \mid x; \varphi) \quad \forall x \in \mathcal{X} \tag{1}$$

states that the deployed model is behaviorally identical to the declared one.

$$H_1: \quad \exists x \in \mathcal{X} \text{ such that } \pi_{\text{tgt}}(\cdot \mid x; \varphi) \neq \pi_{\text{ref}}(\cdot \mid x; \varphi) \tag{2}$$

captures any observable deviation, whether arising from quantization, fine-tuning, prompt injection, or full model replacement.

**Adversarial model provider.** A dishonest model provider may attempt to evade detection through *probabilistic substitution*. We model this behaviour by

$$\pi_{\text{tgt}}(\,\cdot \mid x; \varphi) \;=\; \big(1 - q(x)\big)\, \pi_{\text{ref}}(\,\cdot \mid x; \varphi) \;+\; q(x)\, \pi_{\text{alt}}(\,\cdot \mid x; \varphi), \tag{3}$$

where $\pi_{\text{alt}}$ is an arbitrary alternative model (e.g., quantized or adversarially finetuned version of $\pi_{\text{ref}}$) and $q : \mathcal{X} \to [0, 1]$ is an unknown routing function that may depend on the prompt and even on past interaction history (e.g., routing based on prompt caching (Gu et al., 2025)). When $q(x)$ is small but positive on a non-negligible fraction of typical prompts, the mixture in equation 3 significantly complicates discrimination; nevertheless, a sound detection procedure must retain statistical power under such adversarial conditions.

**Detector and objectives.** A *detector* is a decision rule $\varphi : \mathcal{D}_N \to \{0, 1\}$ whose output 1 means "reject $H_0$". Here, $\mathcal{D}_N$ denotes the data collected with $N$ API queries and many local model queries. For a user-specified significance level $\alpha \in (0, 1)$, the detector must satisfy the type-I error constraint $\Pr_{H_0}\big[\varphi(\mathcal{D}_N) = 1\big] \leq \alpha$. Subject to this guarantee, the verifier seeks to maximize the statistical power

$$\beta(q, \pi_{\text{alt}}) \;=\; \Pr_{H_1}\big[\varphi(\mathcal{D}_N) = 1\big],$$

uniformly over a broad class of substitution policies $(q, \pi_{\text{alt}})$; to do so with as few costly API queries $N$ as possible; and to employ prompts indistinguishable from ordinary user traffic, thereby preventing the provider from detecting and rerouting "suspicious-looking" inputs.

## 4 METHOD

We introduce our rank-based uniformity test in Section 4.1 and empirically identify the most discriminative score function to maximize the detection power in Section 4.2.

### 4.1 RANK-BASED UNIFORMITY TEST (RUT)

To determine whether the target model $\pi_{\text{tgt}}$ is identical to the reference model $\pi_{\text{ref}}$, we propose a *rank-based uniformity test*. This method assesses how typical the target model's responses are under the reference model's generation distribution.

**Response score function.** To perform a statistical test, we first define a scalar-valued *score function* $f : \mathcal{Y} \times \mathcal{X} \to \mathbb{R}$ that maps a model response and prompt to a real number. This function assigns a score to each output given the prompt, i.e.,

$$s = f(y, x), \quad \text{where } x \in \mathcal{X},\ y \in \mathcal{Y},\ s \in \mathbb{R}.$$

An ideal score function $f^*$ should induce an *injective* mapping $y \mapsto f^*(y, x)$ for any fixed prompt $x \in \mathcal{X}$. Under this assumption, each distinct response corresponds to a unique score value, ensuring that the score distribution fully characterizes the model's outputs.

**Uniformity as a test signal.** For each prompt $x \in \mathcal{X}$, we sample a response $y_{\text{tgt}} \sim \pi_{\text{tgt}}(\,\cdot \mid x; \varphi)$ and compute its scalar score $s_{\text{tgt}} = f(y_{\text{tgt}}, x)$. To assess how typical this response is under the reference model, we evaluate its rank in the reference model's score distribution.

We define the cumulative distribution function (CDF) of the reference model's scores as:

$$F_{\pi_{\text{ref}}}(s \mid x) := \mathbb{P}_{y \sim \pi_{\text{ref}}(\cdot \mid x; \varphi)}\left[f(y, x) \leq s\right].$$

Since $f(y, x)$ takes values in a discrete set, $F_{\pi_{\text{ref}}}$ is a step function. To ensure the rank statistic is continuously distributed under the null hypothesis, we apply a *randomized quantile residual* (Dunn & Smyth, 1996) to extend the probability integral transform (David & Johnson, 1948) to discrete distributions. Specifically, we define the *rank statistic* as

$$r_{\text{tgt}} := F_{\pi_{\text{ref}}}(s_{\text{tgt}}^-) + U \cdot \mathbb{P}\left(f(y, x) = s_{\text{tgt}}\right), \quad U \sim \text{Uniform}[0, 1], \tag{4}$$

where $F_{\pi_{\text{ref}}}(s_{\text{tgt}}^-) := \mathbb{P}\left(f(y, x) < s_{\text{tgt}}\right)$ is the left-limit of the CDF at $s_{\text{tgt}}$, and $\mathbb{P}(f(y, x) = s_{\text{tgt}})$ is the probability mass at $s_{\text{tgt}}$. Under the null hypothesis $\pi_{\text{tgt}} = \pi_{\text{ref}}$, this rank statistic $r_{\text{tgt}} \in [0, 1]$ is uniformly distributed.

Conversely, suppose that $r_{\text{tgt}} \sim \text{Uniform}[0, 1]$ under the randomized quantile residual construction. Since the CDF $F_{\pi_{\text{ref}}}(\cdot \mid x)$ is stepwise and non-decreasing, a uniformly distributed $r_{\text{tgt}}$ implies that the score $s_{\text{tgt}}$ follows the same discrete distribution as $s_{\text{ref}}$. By injectivity of $f$, this further implies that $y_{\text{tgt}} \sim \pi_{\text{ref}}(\cdot \mid x; \varphi)$, and hence $\pi_{\text{tgt}} = \pi_{\text{ref}}$.

Thus, with an injective score function $f$, testing the uniformity of $r_{\text{tgt}}$ as defined in equation 4 offers a valid signal for distinguishing $\pi_{\text{tgt}}$ from $\pi_{\text{ref}}$.

**Empirical approximation of $F_{\pi_{\text{ref}}}$.** In practice, it is intractable to build the true CDF $F_{\pi_{\text{ref}}}(\cdot \mid x)$. Instead, we approximate it using an empirical CDF from $m$ reference samples for each prompt.

Given a target response $y_i \sim \pi_{\text{tgt}}(\cdot \mid x_i; \theta)$ and reference responses $y_{ij} \sim \pi_{\text{ref}}(\cdot \mid x_i; \theta)$ for $j = 1, \ldots, m$, we compute the scalar scores

$$s_i := f(y_i, x_i), \quad s_{ij} := f(y_{ij}, x_i).$$

We then define the *randomized rank statistics* $r_i \in [0, 1]$ as

$$r_i = \frac{1}{m} \left( \sum_{j=1}^{m} \mathbf{1}\{s_i > s_{ij}\} + U_i \cdot \sum_{j=1}^{m} \mathbf{1}\{s_i = s_{ij}\} \right),$$

where $U_i \sim \text{Uniform}[0, 1]$ is an independent random variable to break ties uniformly, and ensure $r_i$ is an unbiased estimator of $r_{\text{tgt}}$ given the prompt $x_i$.

**Discriminative score function via empirical selection.** While an ideal *injective* score function would guarantee sensitivity to any difference between $\pi_{\text{tgt}}$ and $\pi_{\text{ref}}$, constructing such a function for which we can calculate the CDF is generally infeasible in practice.

To ensure that our test remains practically effective, we instead require the score function to be *sufficiently discriminative*, in the sense that it induces distinct score distributions whenever $\pi_{\text{ref}} \neq \pi_{\text{tgt}}$. Formally, for fixed prompt $x \in \mathcal{X}$, let

$$S_{\pi_{\text{ref}}} := f(y, x) \text{ with } y \sim \pi_{\text{ref}}(\cdot \mid x; \varphi), \quad \text{and} \quad S_{\pi_{\text{tgt}}} := f(y, x) \text{ with } y \sim \pi_{\text{tgt}}(\cdot \mid x; \varphi).$$

We say that $f$ is sufficiently discriminative if the distributions of $S_{\pi_{\text{ref}}}$ and $S_{\pi_{\text{tgt}}}$ differ whenever $\pi_{\text{ref}} \neq \pi_{\text{tgt}}$, i.e.,

$$\pi_{\text{ref}}(\cdot \mid x; \varphi) \neq \pi_{\text{tgt}}(\cdot \mid x; \varphi) \quad \Rightarrow \quad P_{S_{\pi_{\text{ref}}}} \neq P_{S_{\pi_{\text{tgt}}}}.$$

Under this condition, differences in response distributions are reflected in the score distributions, causing the ranks to deviate from uniformity.

Thus, we aim to find the most discriminative score function among several promising candidates through empirical experiments. In Section 4.2, we compare five candidate score functions—log-likelihood, token rank, log-rank, entropy, and the log-likelihood log-rank ratio (Su et al., 2023)—and find that log-rank is the most discriminative in practice for separating responses by $\pi_{\text{ref}}$ and $\pi_{\text{tgt}}$, and therefore adopt it in our uniformity test.

**Full test procedure.** We now present the full RUT procedure.

Let $\{x_1, \ldots, x_n\} \subset \mathcal{X}$ be a set of prompts. For each prompt $x_i$, we sample one response from the target model,

$$y_i \sim \pi_{\text{tgt}}(\cdot \mid x_i; \theta),$$

and $m$ responses from the reference model,

$$y_{ij} \sim \pi_{\text{ref}}(\cdot \mid x_i; \theta), \quad j = 1, \ldots, m.$$

We compute the log-rank scores

$$s_i := f(y_i, x_i), \quad s_{ij} := f(y_{ij}, x_i),$$

and the corresponding randomized rank statistics $\{r_i\}_{i=1}^{n}$.

We apply the Cramér–von Mises (CvM) test (Cramér, 1928) to assess the deviations between $\{r_i\}_{i=1}^{n}$ and $\text{Uniform}[0, 1]$. The test evaluates the null hypothesis

$$H_0 : r_i \sim \text{Uniform}[0, 1] \quad \text{for all } i.$$

The CvM test statistic is defined as

$$\omega^2 = \frac{1}{12n} + \sum_{i=1}^{n}\left(\frac{2i-1}{2n} - r_{(i)}\right)^2,$$

where $r_{(1)} \leq r_{(2)} \leq \cdots \leq r_{(n)}$ are the ordered rank statistics.

To compute the $p$-value, we compare the observed statistic $\omega^2_{\text{obs}}$ to the distribution of the CvM statistic $\omega^2_{\text{null}}$ computed under the null hypothesis. The $p$-value is given by

$$p\text{-value} = \mathbb{P}_{H_0}\left[\omega^2_{\text{null}} \geq \omega^2_{\text{obs}}\right].$$

We reject $H_0$ and conclude that $\pi_{\text{tgt}}$ and $\pi_{\text{ref}}$ are different if $p$-value $< 0.05$.

## 4.2 Score Function Selection

The RUT requires a scalar score function $f(y, x)$. To identify a function that best captures distributional differences between models, we consider five candidate functions. **Log-likelihood:** $\log \pi_{\text{ref}}(y \mid x)$. **Token rank:** the average rank of response tokens in $y$, where a token's rank is its position in the vocabulary ordered by the $\pi_{\text{ref}}$'s next-token probabilities. **Log-rank:** the average of the logarithm of the token rank. **Entropy:** predictive entropy for $y$ under $\pi_{\text{ref}}(x)$. **Log-likelihood log-rank ratio (LRR):** the ratio between log-likelihood and log-rank. (Su et al., 2023).

To identify the most discriminative score function, we conduct a Monte Carlo evaluation consisting of 500 independent trials. In each trial, we randomly select 10 prompts from the Wild-Chat (Zhao et al., 2024) dataset and sample 50 completions per prompt from both $\pi_{\text{ref}}$ and $\pi_{\text{tgt}}$, using a fixed temperature of 0.5 and a maximum length of 30 tokens. For each candidate score function, we compute the average AU-ROC (Bradley, 1997) across the 10 prompts for each trial, yielding a distribution of 500 AU-ROC scores per function. The full algorithm to calculate per score function average AUROC is included in Appendix A.1. Across different model comparisons, we find that **log-rank** consistently yields the most separable AUROC distribution from 0.5, indicating the strongest discriminative power. Figure 2 shows an example

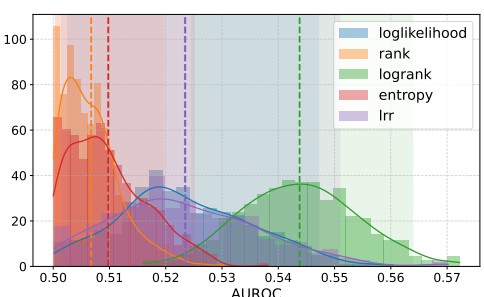

Figure 2: Distribution of AUROC scores for five candidate score functions across 500 trials comparing Gemma-2-9b-it and its 4-bit quantized variant. Log-rank achieves the most separable distribution from the random level 0.5, indicating superior power in distinguishing different models.

comparing Gemma-2-9b-it with its 4-bit quantized variant. Based on the results, we select log-rank as the scoring function for RUT. Complete AUROC results are provided in Appendix A.2. We also present a formal analysis of RUT's statistical power with the log-rank score function under adversarial perturbations in Appendix C.

## 5 Experiments

In this section, we evaluate RUT across diverse model substitution scenarios, including quantization (Section 5.2), jailbreaks (Section 5.3), SFT (Section 5.4), full model replacement (Section 5.5), additional query domains (Section 5.6), and real-world API providers (Section 5.7). Detection performance is compared against MMD and a KS baseline using statistical power AUC as the primary metric. We also include a case study on detecting decoding parameter mismatch in Appendix B.6 and demonstrate that RUT remains robust across models and query domains.

## 5.1 Experimental Setup

To evaluate detection performance under adversarial conditions, we simulate probabilistic substitution attacks where a fraction $q \in [0, 1]$ of API queries are routed to an alternative model (e.g., quantized or fine-tuned). For each value of $q$, we estimate the statistical power, defined as the probability of correctly rejecting the null hypothesis when substitution is present. We then summarize the

resulting power–substitution rate curve using the area under the curve (AUC) over $q \in [0, 1]$. The AUC ranges from 0 to 1 and reflects the method's ability to maintain high statistical power across varying levels of substitution, serving as a measure of robustness to such attacks. Higher values indicate more reliable and consistent detection performance. Figure 1 shows an example comparing Gemma-2-9b-it and its 4-bit quantized variant. In the following experiments, we compute 95% confidence intervals for AUCs using bootstrapping. The 95% wilson confidence intervals for statistical powers are shown as shaded area on plots in Appendix B.

**Data.** We use the WildChat dataset (Zhao et al., 2024), which contains real-world conversations between human users and ChatGPT. This dataset reflects authentic user behavior, ensuring the query distribution represents typical API traffic.

**Baseline.** For the detection methods (Sun et al., 2025; Gao et al., 2025) that are compatible with WildChat, We primarily focus on Maximum Mean Discrepancy (MMD) (Gao et al., 2025) as the baseline, as Sun et al. (2025) is reported to fail to identify quantization (Cai et al., 2025). We also tailor a Kolmogorov–Smirnov (KS) test baseline that uses the same information as RUT: it computes the log-rank scores from the reference model on both the target and reference model responses and applies the two-sample KS test (Darling, 1957) on these two sets of scores to estimate the $p$-value.

**Test procedures.** We apply a consistent sample budget constraint on all tests. The implementation details of their test procedures are listed below:

- **Rank-Based Uniformity Test (RUT):** Each trial samples 100 prompts. We query each prompt once to the target and 100 times to the reference model.
- **Maximum Mean Discrepancy (MMD):** We apply the MMD test based on the character-level Hamming distance following Gao et al. (2025). Each trial uses 10 prompts, with 10 samples per prompt. We compute the MMD statistic and estimate the $p$-value via 500 random permutations.
- **Kolmogorov–Smirnov Test (KS):** We use the same sampling setup as RUT: 100 prompts per trial, 1 query to the target, and 100 to the reference model per prompt.

Across all models, we set the temperature to $0.5$ and cap generation at 30 tokens. We use vLLM (Kwon et al., 2023) on a single A6000 for local inferences. Statistical power is estimated over 500 Monte Carlo trials as the proportion of trials correctly rejecting the null at $\alpha = 0.05$. All tests were run with Intel Xeon Gold 6230R @ 2.10GHz and 16 GB RAM.

## 5.2 DETECTING QUANTIZATION

We consider the setting where the API provider uses a quantized variant to substitute the claimed model. We evaluate three detection methods on quantized variants of Llama-3.2-3B-Instruct[1], Mistral-7B-Instruct-v0.3[2], and Gemma-2-9B-it[3], comparing each model to its 4-bit and 8-bit quantized counterparts. As shown in Table 2a, none of the methods succeed in reliably detecting substitution for the 8-bit variants of Gemma and Mistral, where statistical power AUC remains near zero across the board. In the remaining four settings, RUT outperforms MMD and the KS baseline in three out of the four cases, demonstrating superior sensitivity to quantization-induced distributional shifts. Full statistical power curves for AUCs are provided in Appendix B.1.

## 5.3 DETECTING JAILBREAKS

We consider the setting where the API provider secretly appends a hidden jailbreaking system prompt to user queries. To evaluate this scenario, we use two base models: Mistral-7B-Instruct-v0.3 and Gemma-2-9B-it. For each model, we construct a test using three representative jailbreaking prompts *Dan*, *Anti-Dan*, and *Evil-Bot* adapted from Shen et al. (2024). As shown in Table 2b, all jailbreak cases are reliably detected, with power AUCs consistently above 0.75. RUT achieves the highest power in all 6 settings, demonstrating its superior sensitivity to model deviations caused by hidden jailbreaking prompts. Full statistical power curves for AUCs are provided in Appendix B.2.

## 5.4 DETECTING SFT

We study the setting where the API provider fine-tunes a model on instruction-following data. Specifically, we fine-tune two base models—Llama-3.2-3B-Instruct and Mistral-7B-Instruct-

---

[1]https://huggingface.co/meta-llama/Llama-3.2-3B-Instruct
[2]https://huggingface.co/mistralai/Mistral-7B-Instruct-v0.3
[3]https://huggingface.co/google/gemma-2-9b-it

(a) Statistical power AUC for detecting quantized variants. **Bold** = best method; gray = none reliable.

| Model | RUT | MMD | KS |
|---|---|---|---|
| Gemma–4bit | $0.392_{84^- \ 103^+}$ | $0.214_{110^- \ 112^+}$ | $0.017_{32^- \ 34^+}$ |
| Gemma–8bit | $0.049_{59^- \ 59^+}$ | $0.043_{62^- \ 64^+}$ | $0.001_{08^- \ 10^+}$ |
| Llama–4bit | $\mathbf{0.642}_{84^- \ 82^+}$ | $0.625_{89^- \ 93^+}$ | $0.474_{88^- \ 89^+}$ |
| Llama–8bit | $0.132_{82^- \ 81^+}$ | $\mathbf{0.158}_{110^- \ 109^+}$ | $0.005_{17^- \ 19^+}$ |
| Mistral–4bit | $\mathbf{0.586}_{95^- \ 94^+}$ | $0.500_{92^- \ 97^+}$ | $0.330_{100^- \ 102^+}$ |
| Mistral–8bit | $0.049_{56^- \ 61^+}$ | $0.090_{77^- \ 84^+}$ | $0.006_{20^- \ 22^+}$ |

(b) Statistical power AUC for detecting jail-breaking prompts. **Bold** = most effective method per prompt.

| Model | Prompt | RUT | MMD | KS |
|---|---|---|---|---|
| Mistral | Dan | $\mathbf{0.895}_{46^- \ 45^+}$ | $0.802_{63^- \ 55^+}$ | $0.873_{37^- \ 36^+}$ |
| | Anti-Dan | $\mathbf{0.893}_{43^- \ 41^+}$ | $0.781_{52^- \ 55^+}$ | $0.872_{46^- \ 49^+}$ |
| | Evil-Bot | $\mathbf{0.892}_{44^- \ 44^+}$ | $0.766_{56^- \ 59^+}$ | $0.873_{37^- \ 37^+}$ |
| Gemma | Dan | $\mathbf{0.888}_{43^- \ 44^+}$ | $0.757_{65^- \ 68^+}$ | $0.867_{34^- \ 36^+}$ |
| | Anti-Dan | $\mathbf{0.858}_{49^- \ 53^+}$ | $0.816_{60^- \ 52^+}$ | $0.854_{31^- \ 31^+}$ |
| | Evil-Bot | $\mathbf{0.893}_{43^- \ 42^+}$ | $0.753_{65^- \ 63^+}$ | $0.871_{36^- \ 36^+}$ |

Table 2: Statistical power AUCs. All confidence interval ranges are reported in $\times 10^{-4}$

v0.3—on benign and harmful instruction-following datasets. We use Alpaca (Taori et al., 2023) as the benign dataset and BeaverTails (Ji et al., 2023) for harmful question answering. Each model is fine-tuned on 500 samples from the respective dataset for 5 epochs using LoRA (Hu et al., 2021) with rank 64 and $\alpha = 16$, a batch size of 32, and a learning rate of $1 \times 10^{-4}$ on a single A100. For each checkpoint, we compute the statistical power AUC of the detection methods.

As shown in Figure 3, RUT consistently achieves higher power AUC than both the KS and MMD baselines across all fine-tuning configurations. Notably, our method detects behavioral changes within the first epoch of fine-tuning, demonstrating strong sensitivity to early-stage distributional shifts. While all methods improve with additional training, RUT remains the most robust across both models and datasets. Full statistical power curves for AUCs are provided in Appendix B.3.

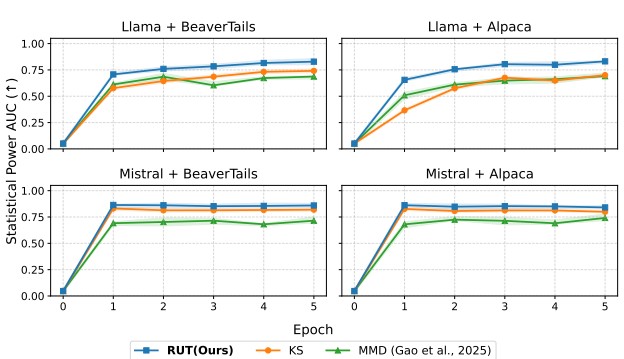

Figure 3: AUC of SFT checkpoints across epochs.

## 5.5 DETECTING FULL MODEL REPLACEMENT

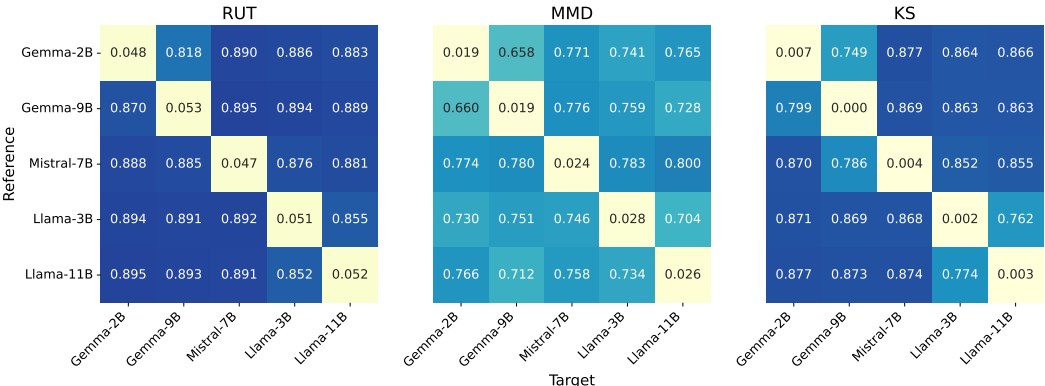

Figure 4: AUC for detecting full model replacement. Each cell shows the AUC score between a reference and a target model. Diagonal values represent self-comparisons.

We evaluate the setting where the API provider substitutes the claimed model with a completely different one. To simulate this scenario, we conduct pairwise comparisons among five open-

source models: Llama-3.2-3B-Instruct, Llama-3.2-11B-Vision-Instruct[4], Mistral-7B-Instruct-v0.3, Gemma-2-2B-it[5], and Gemma-2-9B-it. For each pair, one model serves as the reference model while the other acts as the deployed target model. As shown in Figure 4, RUT consistently achieves the highest statistical power AUC across model pairs, outperforming both the MMD and KS baselines. The results highlight the method's sensitivity to full model substitutions. Full statistical power curves for AUCs are provided in Appendix B.4.

## 5.6 DETECTING QUERY DOMAINS

We now evaluate the robustness of RUT across more query domains. Beyond WildChat which reflects general conversational traffic, we consider two specialized datasets: BigCodeBench (Zhuo et al., 2024) for programming tasks and MATH (Hendrycks et al., 2021) for mathematical problem solving. We adopt the quantization setup in Section 5.2 as this setting is both challenging and practically relevant. See full power curves in Appendix B.5.

As shown in Table 3, the overall detection powers mirror those observed in Section 5.2. Detection remains difficult for 8-bit quantized Gemma and Mistral, where all methods fail to achieve meaningful power. RUT consistently shows high detectability in the remaining cases, outperforming MMD and KS in seven out of the eight cases. These results show that RUT remains effective in both math and code domains, reinforcing its generalizability to diverse query distributions.

Table 3: Statistical power AUC for detecting quantized variants across query domains. **Bold** = best method; gray = none reliable. All confidence interval ranges are reported in $\times 10^{-4}$

(a) BigCodeBench (Zhuo et al., 2024)

| Model | RUT | MMD | KS |
|---|---|---|---|
| Gemma–4bit | $0.136$ $_{98^-\ 95^+}$ | $\mathbf{0.170}$ $_{89^-\ 103^+}$ | $0.000$ $_{22^-\ 12^+}$ |
| Gemma–8bit | $0.048$ $_{62^-\ 71^+}$ | $0.052$ $_{69^-\ 58^+}$ | $0.000$ $_{05^-\ 09^+}$ |
| Llama–4bit | $\mathbf{0.593}$ $_{54^-\ 50^+}$ | $0.353$ $_{81^-\ 80^+}$ | $0.524$ $_{48^-\ 50^+}$ |
| Llama–8bit | $0.093$ $_{34^-\ 33^+}$ | $\mathbf{0.143}$ $_{56^-\ 58^+}$ | $0.021$ $_{12^-\ 23^+}$ |
| Mistral–4bit | $\mathbf{0.462}$ $_{76^-\ 84^+}$ | $0.326$ $_{91^-\ 88^+}$ | $0.183$ $_{43^-\ 51^+}$ |
| Mistral–8bit | $0.041$ $_{73^-\ 74^+}$ | $0.058$ $_{67^-\ 69^+}$ | $0.011$ $_{77^-\ 83^+}$ |

(b) Math (Hendrycks et al., 2021)

| Model | RUT | MMD | KS |
|---|---|---|---|
| Gemma–4bit | $\mathbf{0.297}$ $_{102^-\ 133^+}$ | $0.237$ $_{99^-\ 105^+}$ | $0.059$ $_{76^-\ 75^+}$ |
| Gemma–8bit | $0.048$ $_{64^-\ 64^+}$ | $0.067$ $_{72^-\ 75^+}$ | $0.002$ $_{33^-\ 35^+}$ |
| Llama–4bit | $\mathbf{0.532}$ $_{99^-\ 94^+}$ | $0.468$ $_{101^-\ 85^+}$ | $0.412$ $_{45^-\ 44^+}$ |
| Llama–8bit | $\mathbf{0.253}$ $_{72^-\ 73^+}$ | $0.177$ $_{65^-\ 66^+}$ | $0.118$ $_{75^-\ 80^+}$ |
| Mistral–4bit | $\mathbf{0.360}$ $_{33^-\ 37^+}$ | $0.216$ $_{66^-\ 65^+}$ | $0.160$ $_{33^-\ 34^+}$ |
| Mistral–8bit | $0.059$ $_{44^-\ 47^+}$ | $0.060$ $_{58^-\ 58^+}$ | $0.019$ $_{45^-\ 43^+}$ |

## 5.7 DETECTING REAL API PROVIDERS

We evaluate three models—Llama-3.2-3B, Mistral-7B, and Gemma-2-9B—across multiple API providers, using local A100 inference as the baseline. As shown in Table 4, all tests correctly confirm behavioral equivalence in local deployments.

Across all settings, RUT and MMD generally agree in detecting significant deviations across providers, offering mutual validation for their behavioral sensitivity. The KS test exhibits similar trends but with notably lower sensitivity. One exception is Mistral + HF Inference, where MMD yields a power of 1.0 while others remain below 0.2. We trace this to a tokenization mismatch: the HF Inference API consistently omits the leading whitespace present in the reference outputs. Since MMD uses string Hamming distance, the formatting

Table 4: Statistical power for detecting differences from the model deployed on an A6000 GPU. `A100` denotes the same model on an A100 GPU. Values > 0.5 indicate significant behavioral deviation. Green = no significant difference; Red = significant difference.

| Model | Provider | RUT | MMD | KS |
|---|---|---|---|---|
| Llama | A100 | 0.094 | 0.142 | 0.002 |
| Llama | Nebius | 0.962 | 0.944 | 0.426 |
| Llama | Novita | 0.988 | 0.996 | 0.530 |
| Mistral | A100 | 0.058 | 0.138 | 0.004 |
| Mistral | HF Inf. | 0.188 | 1.00 | 0.000 |
| Gemma | A100 | 0.060 | 0.084 | 0.000 |
| Gemma | Nebius | 0.312 | 0.432 | 0.008 |

---

[4] https://huggingface.co/meta-llama/Llama-3.2-11B-Vision-Instruct
[5] https://huggingface.co/google/gemma-2-2b-it

difference inflates the score. After restoring the missing space, the MMD score drops to 0.211, aligning with other tests. This shows the robustness of RUT to minor decoding mismatches that can mislead string-based metrics.

## 6 CONCLUSION

The stable increase in the size (Kaplan et al., 2020) and architectural complexity (Zhou et al., 2022) of frontier LLMs has led to a rise in the popularity of API-based model access. To prevent performance degradation and security risks from model substitution behind API interfaces, this work proposes the rank-based uniformity test for model equality testing. We test the method against a variety of different substitution attacks and demonstrate its consistent effectiveness in detecting substitution and its superiority over existing methods.

**Limitations and Future Work**  We have not empirically validated effectiveness of RUT against an adversary with full knowledge of the auditing method. For example, an attacker could selectively reroute prompts that are expected to produce atypical log-rank statistics. Assessing the method's robustness against more powerful adversaries is an important next step. In addition, RUT requires access to a locally deployed authentic reference model, which limits its applicability to open-sourced models. Exploring ways to relax this requirement would broaden the method's applicability.

By developing an effective and stealthy API-based test for model equality, we hope to advance the safety and security of LLM-based applications in the age of increasingly cloud-based deployment.

## ACKNOWLEDGEMENTS

Thanks Muru Zhang and Oliver Liu for valuable discussions. The work was supported in part by the National Science Foundation (NSF) under Award No. 2427856. Any opinions, findings, and conclusions or recommendations expressed in this material are those of the author(s) and do not necessarily reflect the views of NSF.

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

# A  AUROC

## A.1  AUROC ALGORITHM

---

**Algorithm 1:** Average AUROC for score function evaluation

---

**Input:** Prompt set $\mathcal{D}$; models $\pi_{\text{ref}}$, $\pi_{\text{tgt}}$; decoding parameters $\varphi = (\tau, L)$, where $\tau$ is temperature and $L$ is the maximum generation length; number of prompts $n$; number of completions per prompt per model $m$; score functions $\{\varphi_1, \ldots, \varphi_K\}$.

**Output:** Mean AUROC per score function, denoted $\mu_{\text{AUROC}}(\delta)$.

1  Draw $\{x_1, \ldots, x_n\} \sim \text{Uniform}(\mathcal{D})$;

2  **for** $i \in \{1, \ldots, n\}$ **do**

3  $\quad \{y_{\text{ref}}^{(j)}\}_{j=1}^{m} \sim \pi_{\text{ref}}(\cdot \mid x_i; \varphi)$;

4  $\quad \{y_{\text{tgt}}^{(j)}\}_{j=1}^{m} \sim \pi_{\text{tgt}}(\cdot \mid x_i; \varphi)$;

5  $\quad \mathcal{Y}_i \leftarrow \{y_{\text{ref}}^{(j)}\} \cup \{y_{\text{tgt}}^{(j)}\}$;

6  $\quad L_i \leftarrow \{0\}^m \cup \{1\}^m$;

7  $\quad$ **for** $\delta \in \{\varphi_1, \ldots, \varphi_K\}$ **do**

8  $\quad\quad S_i \leftarrow \{\delta(y) \mid y \in \mathcal{Y}_i\}$;

9  $\quad\quad$ Store $A_i^{\delta} \leftarrow \text{AUROC}(S_i, L_i)$;

10  **for** $\delta \in \{\varphi_1, \ldots, \varphi_K\}$ **do**

11  $\quad \mu_{\text{AUROC}}(\delta) \leftarrow \frac{1}{n} \sum_{i=1}^{n} A_i^{\delta}$;

---

*Note.* $\text{AUROC}(S, L)$ denotes the standard binary AUROC (Bradley, 1997).

## A.2  AUROC SCORE DISTRIBUTIONS

We present the AUROC score distributions from the score function selection experiment described in Section 4.2. Specifically, we evaluated Gemma-2-9B-it, LLaMA-3.2-3B-Instruct, and Mistral-7B-Instruct, and visualized the distributions when distinguishing the original model outputs from three types of variants: (1) quantized versions, (2) models subjected to jailbreaking prompts, and (3) models served by A100 or external API providers.

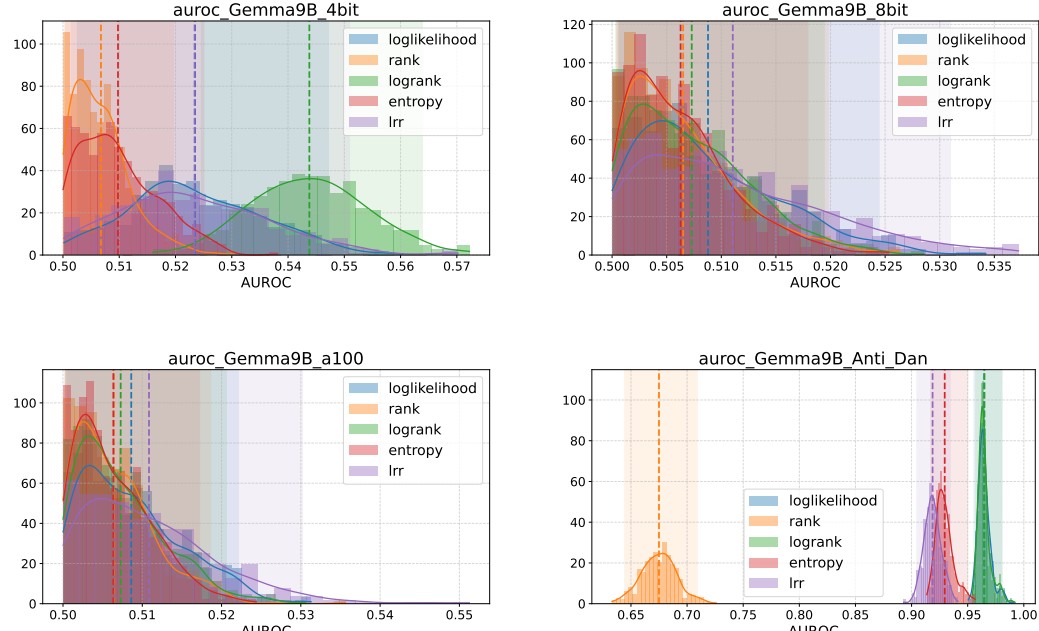

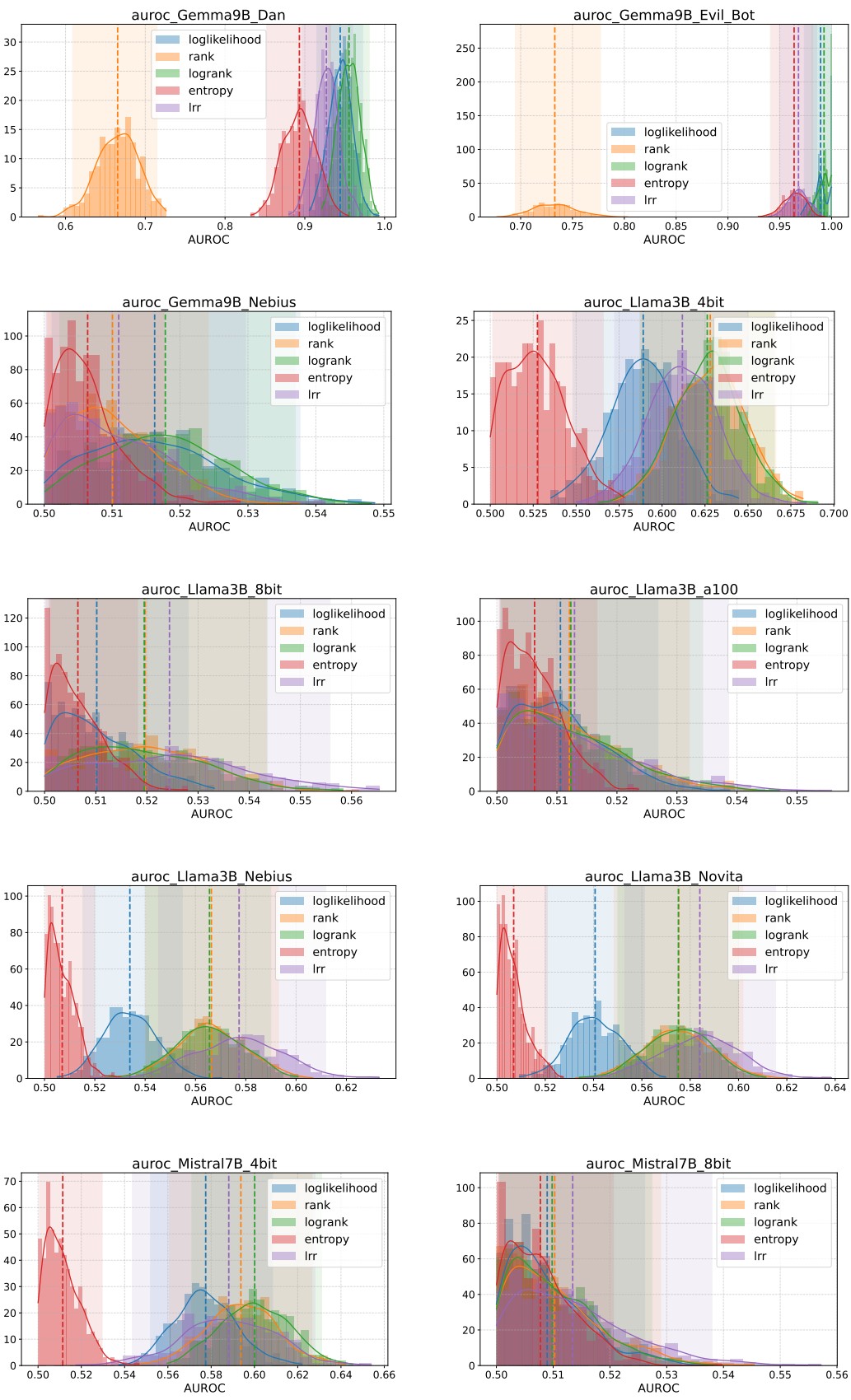

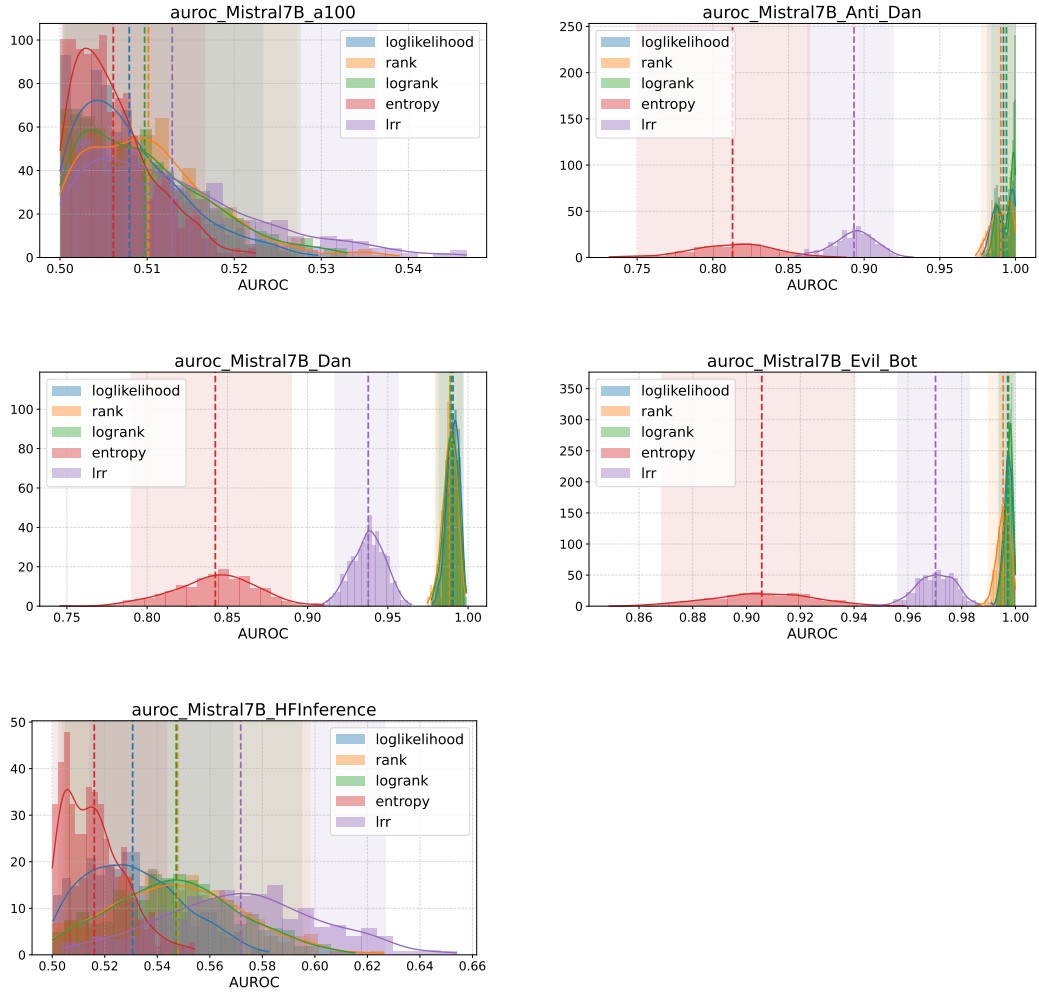

# B    STATISTIC POWER CURVES

## B.1    FULL STATISTIC POWER CURVES FOR DETECTING QUANTIZATION

We present the full statistical power curves, showing the relationship between substitution rate and detection power, corresponding to the experiments on detecting quantized model substitutions described in Section 5.2. These curves are used to compute the power AUC values reported in the main paper and illustrate each method's detection power across different levels of substitution.

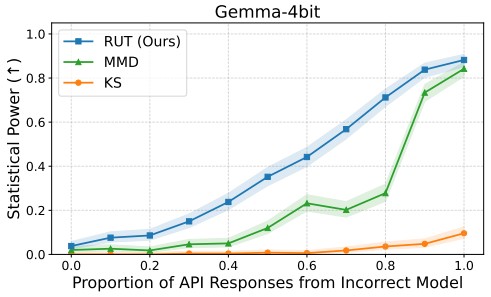
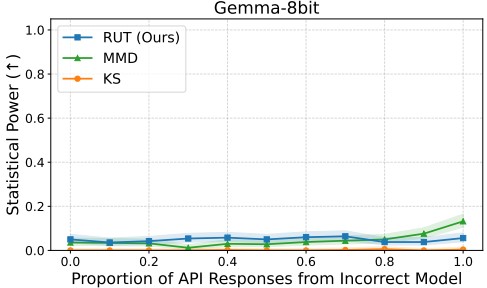

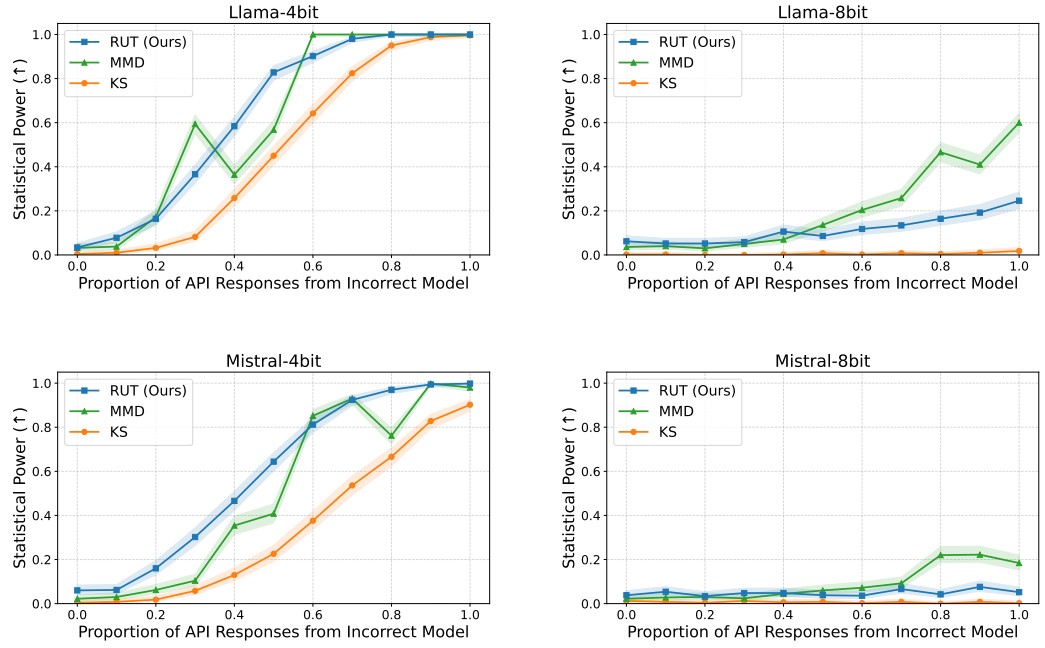

## B.2 Full Statistic Power Curves for Detecting Jailbreaking

We present the full statistical power curves, showing the relationship between substitution rate and detection power, corresponding to the experiments on detecting jailbreak prompts described in Section 5.3. These curves are used to compute the power AUC values reported in the main paper and illustrate each method's detection power across different levels of substitution.

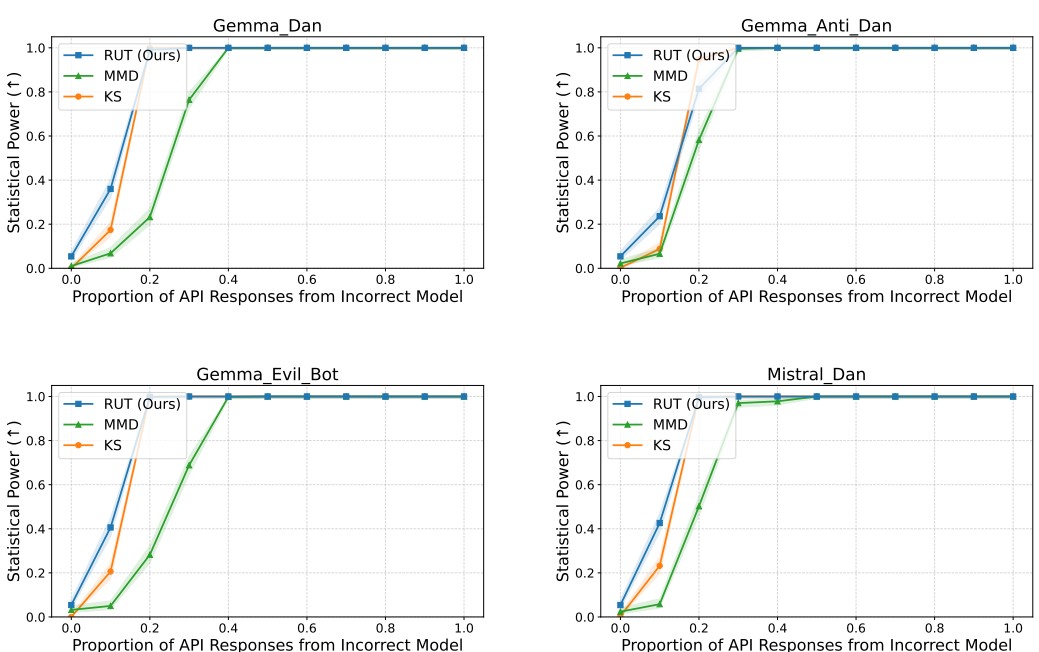

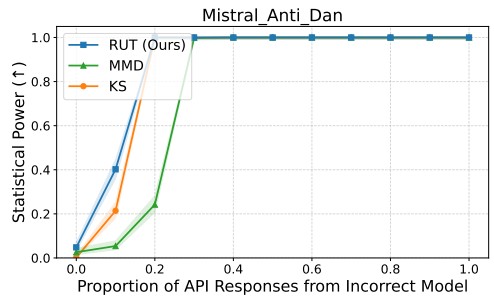
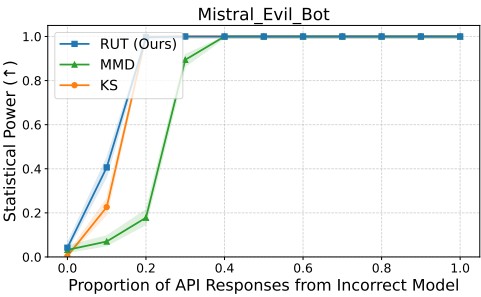

## B.3 FULL STATISTIC POWER CURVES FOR DETECTING SFT

We present the full statistical power curves, showing the relationship between substitution rate and detection power, corresponding to the experiments on detecting SFT described in Section 5.4. These curves are used to compute the power AUC values reported in the main paper and illustrate each method's detection power across different levels of substitution.

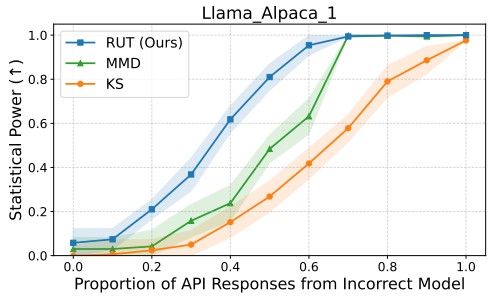
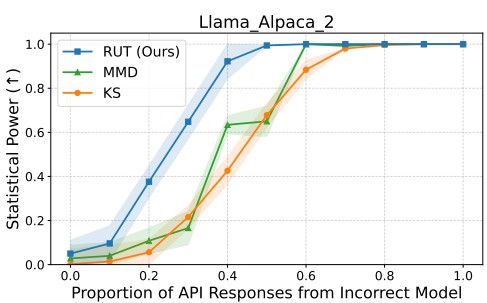

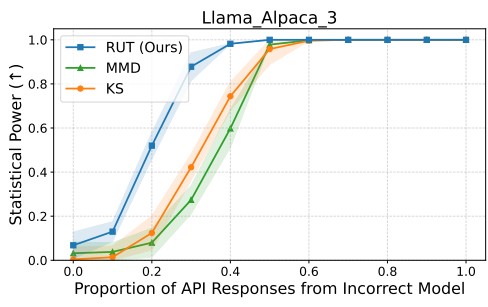
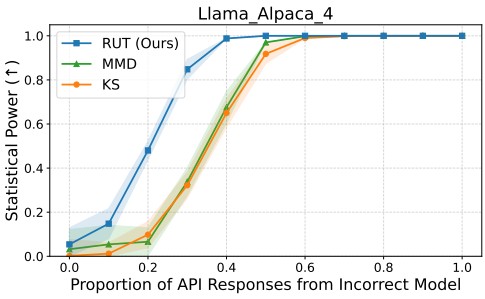

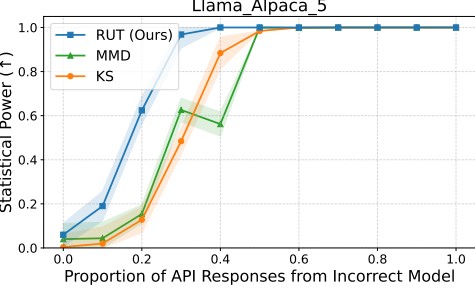

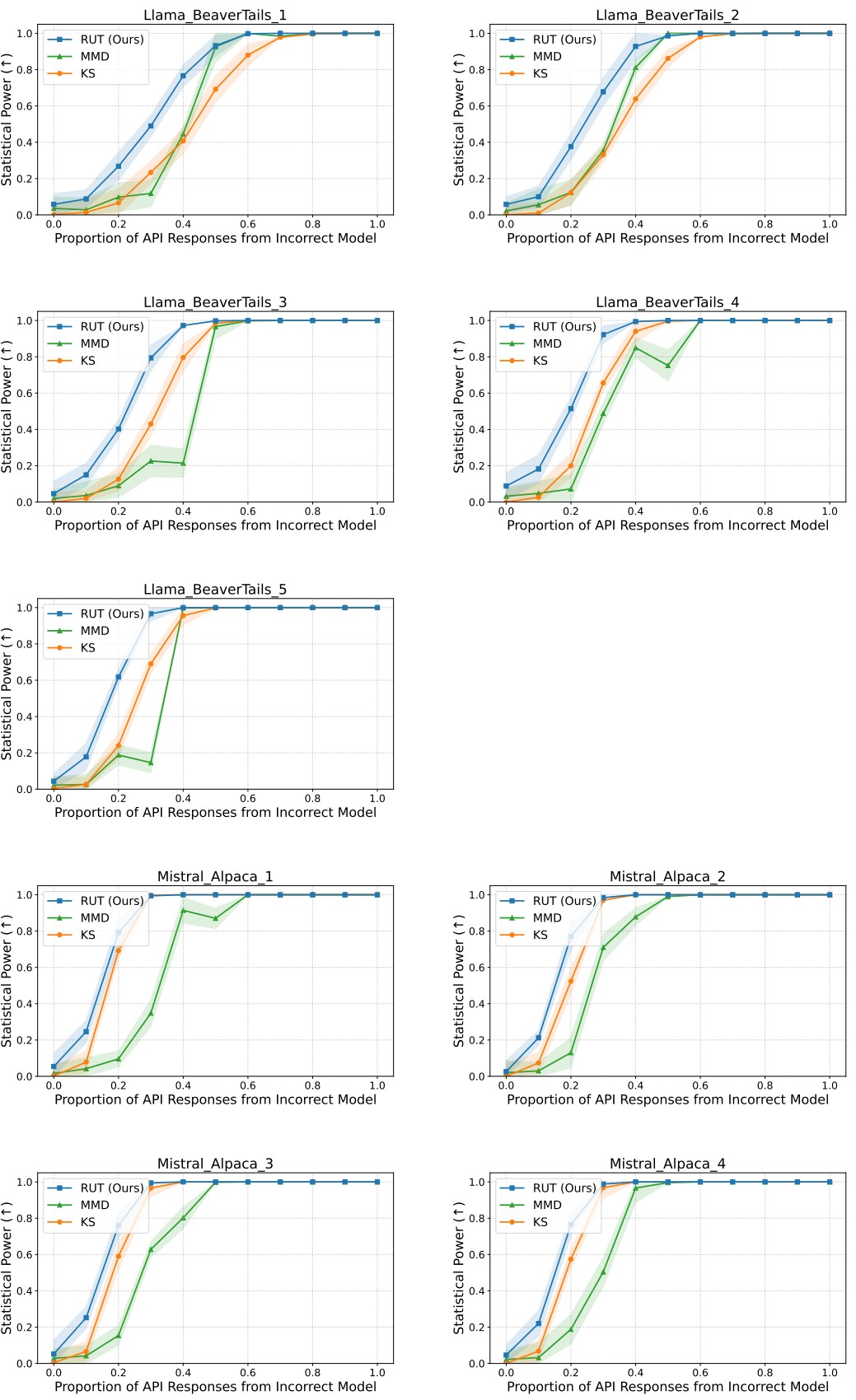

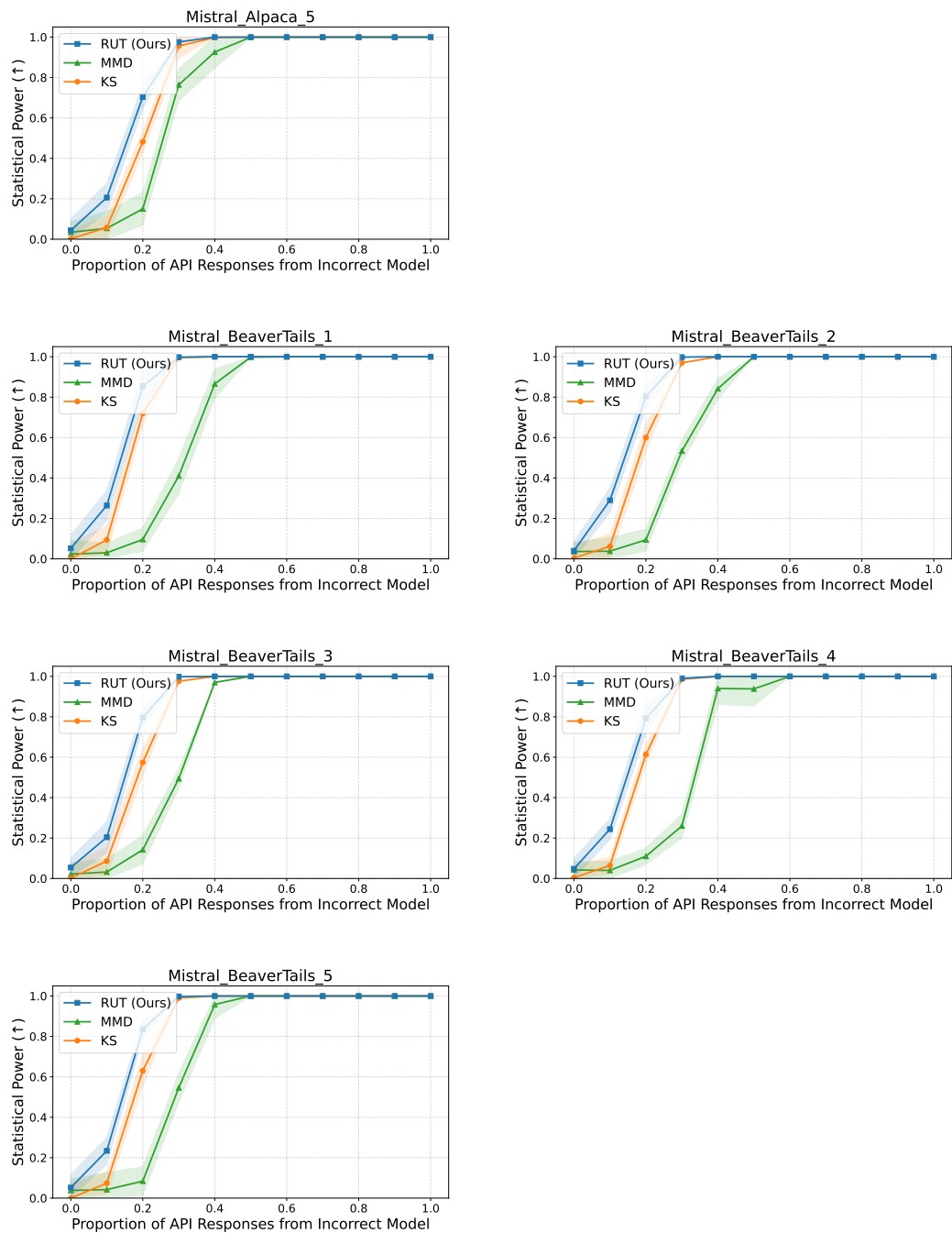

## B.4 FULL STATISTIC POWER CURVES FOR DETECTING FULL MODEL REPLACEMENT

We present the full statistical power curves, showing the relationship between substitution rate and detection power, corresponding to the experiments on detecting full model replacements described in Section 5.5. These curves are used to compute the power AUC values reported in the main paper and illustrate each method's detection power across different levels of substitution.

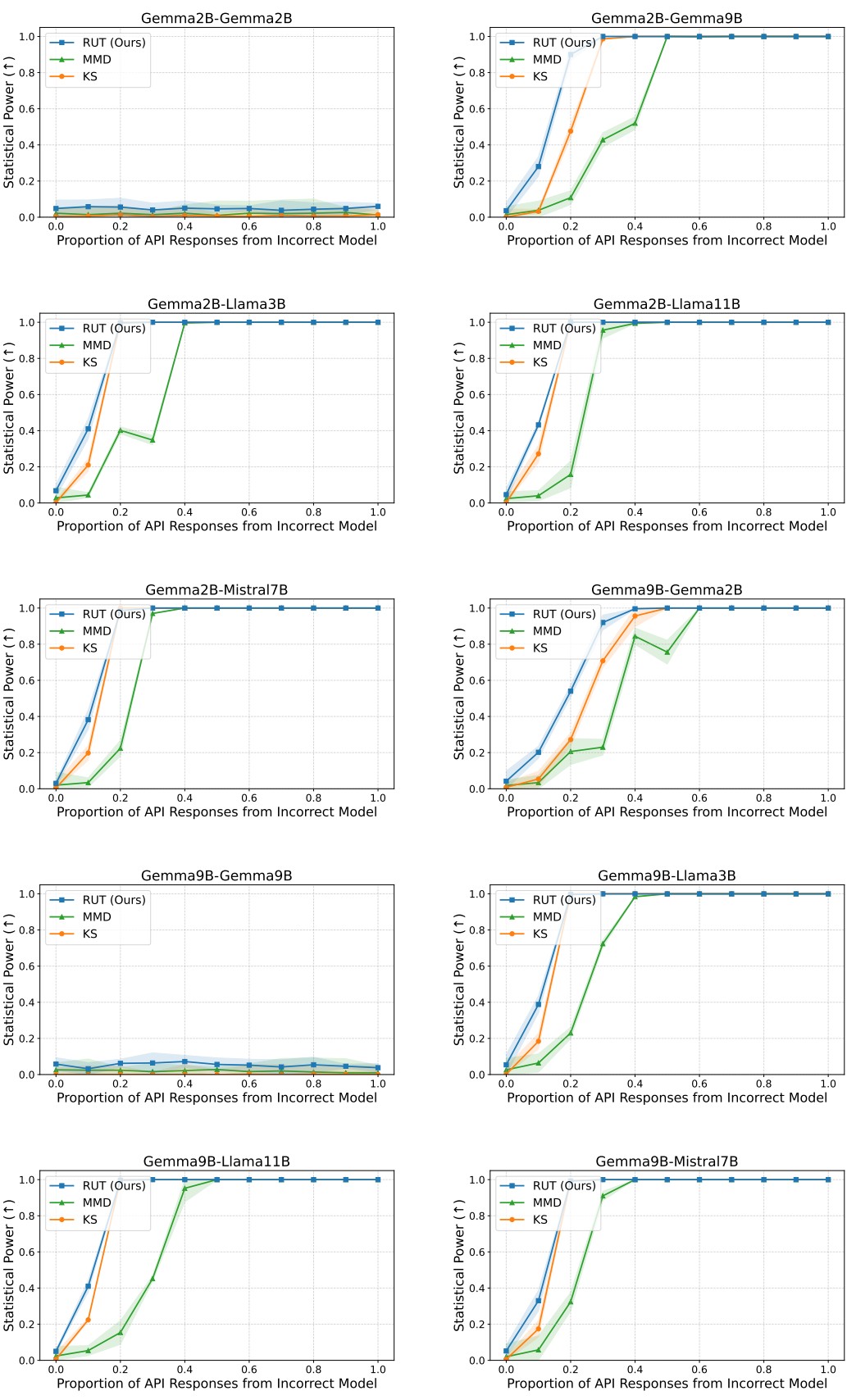

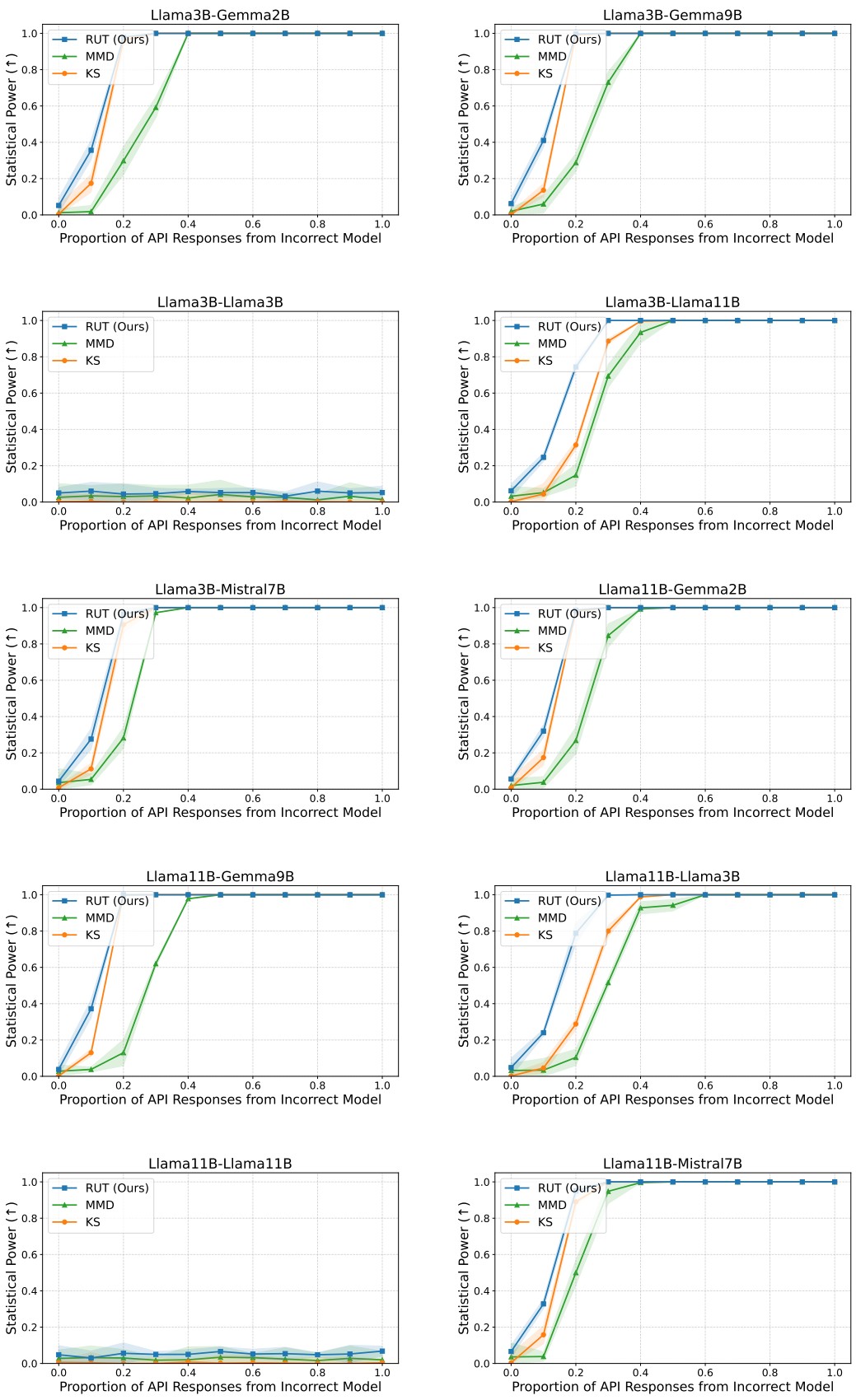

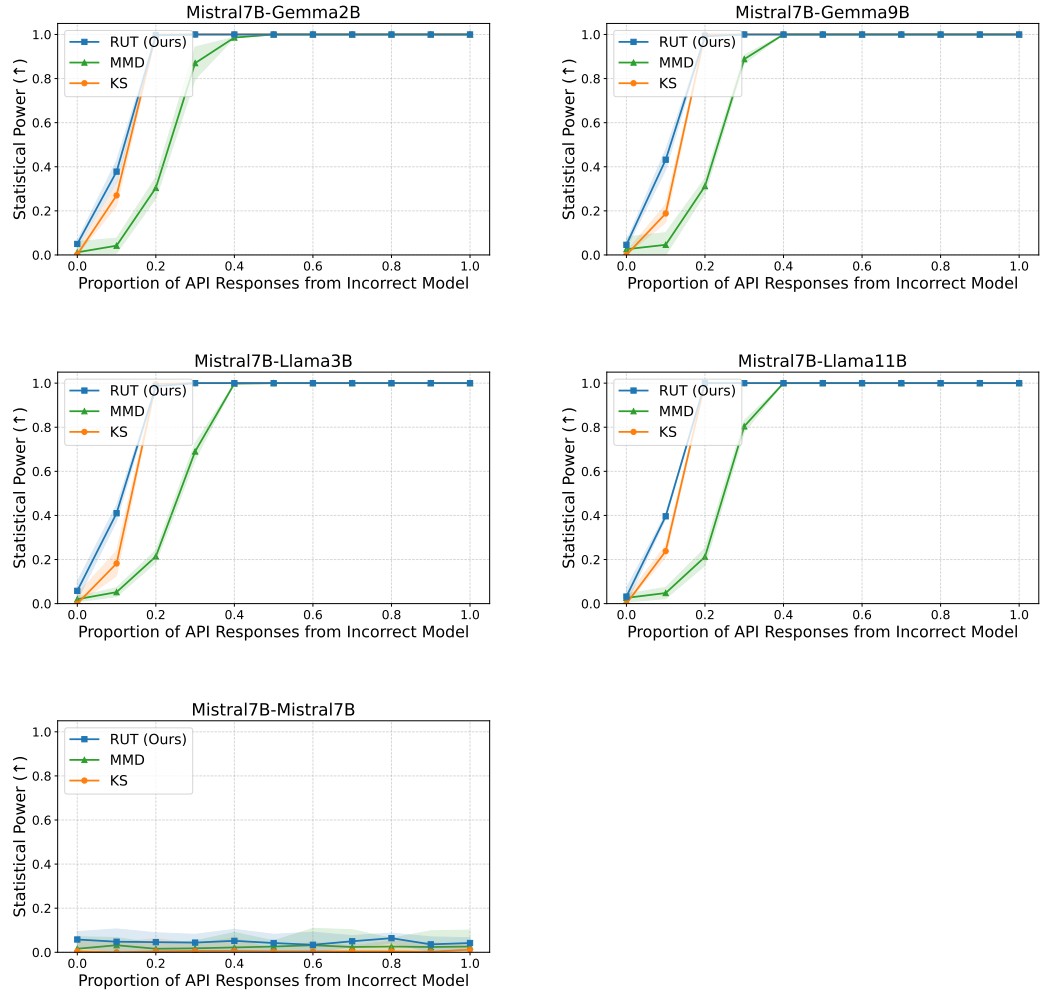

## B.5 Full Statistic Power Curves for Detecting More Query Domains

We present the full statistical power curves, showing the relationship between substitution rate and detection power, corresponding to the experiments on detecting quantized model in extra domains described in Section 5.6. These curves are used to compute the power AUC values reported in the main paper and illustrate each method's detection power across different levels of substitution.

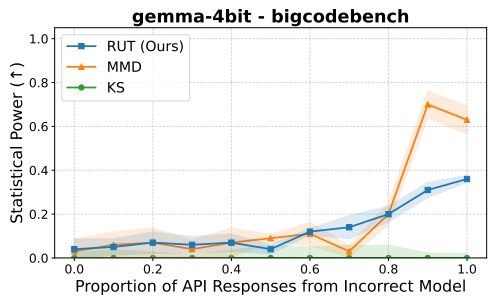

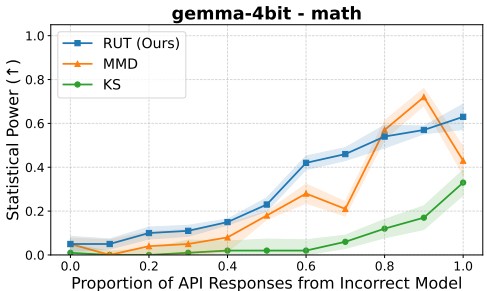

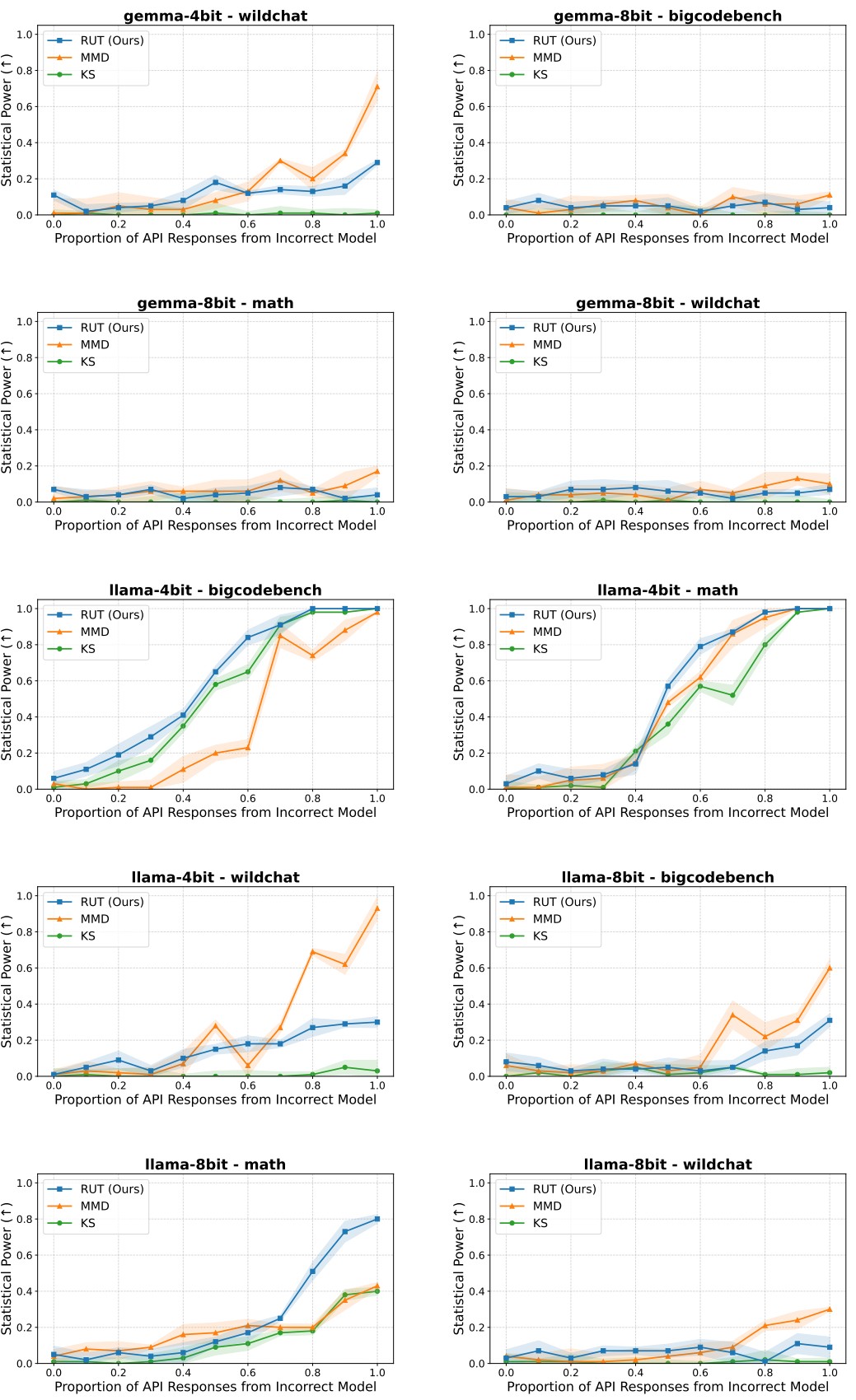

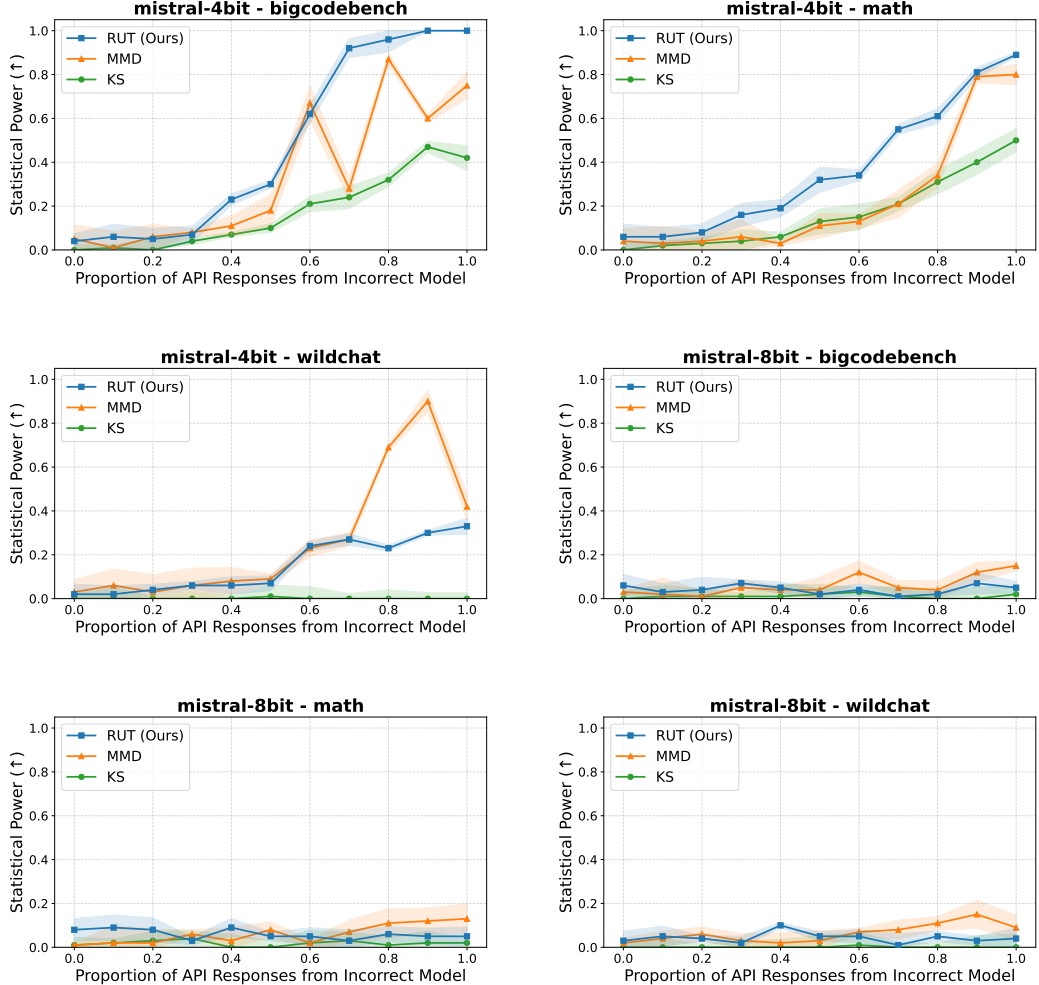

## B.6 CASE STUDY ON DETECTING DECODING PARAMETERS

**Setup.** As a case study, we test whether detection methods can identify changes in decoding parameters, focusing on sampling temperature and top-$p$ for nucleus sampling (Holtzman et al., 2020). We compare responses generated under different parameter settings against the default configuration of temperature=0.5 and top-$p$=1.

**Findings.** We perform the experiments across Gemma-2-9B-it and Llama-3.2-3B-Instruct on MATH (Hendrycks et al., 2021) and WildChat (Zhao et al., 2024). Based on the results in Figure 5, RUT achieves consistently higher detection power across decoding configurations compared to MMD and KS. This sensitivity is desirable in practice, since when the API providers expose decoding controls to users, a reliable detection method should be able to flag deviations arising not only from model substitution but also from anomalous decoding configurations.

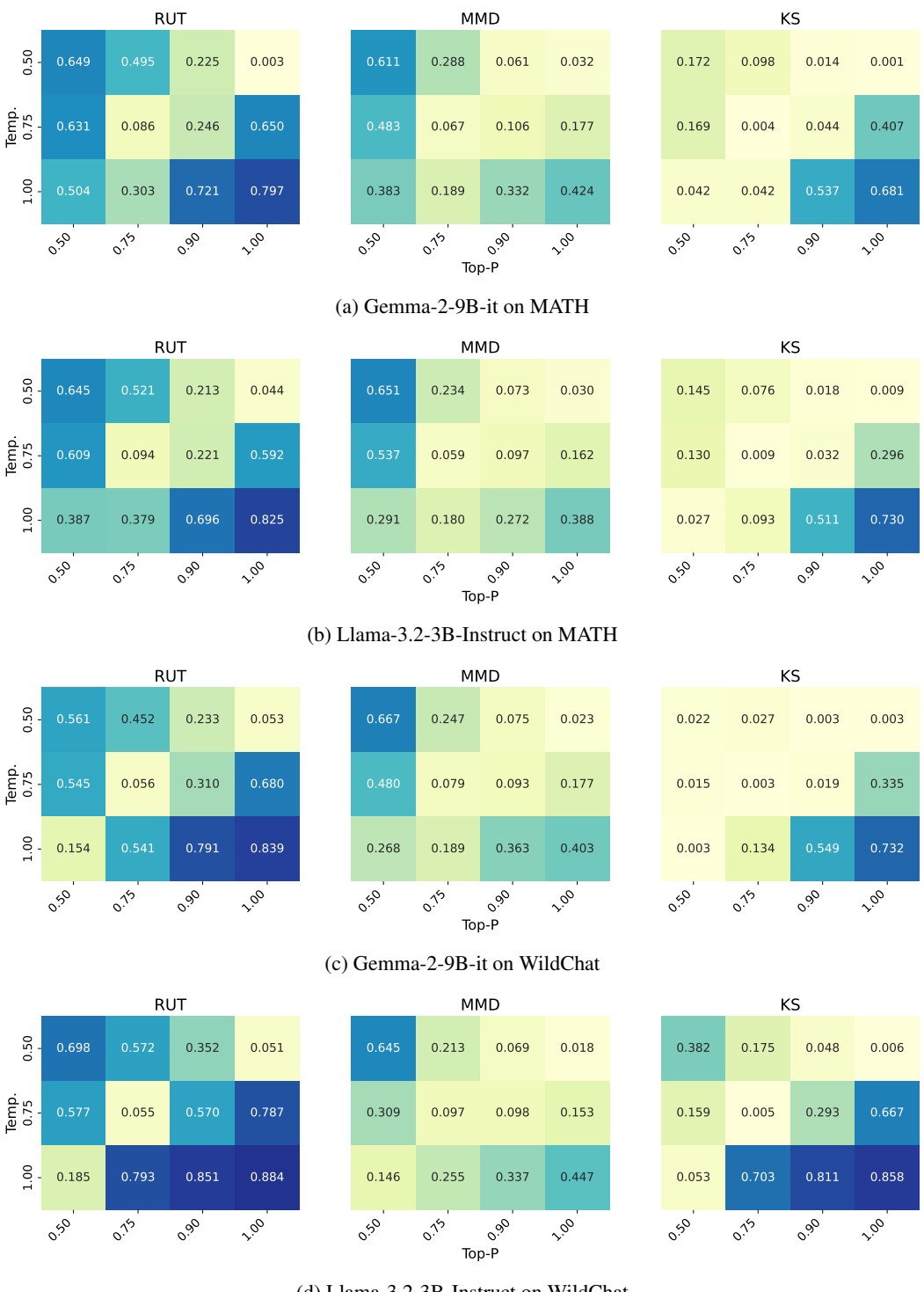

Figure 5: Statistical power AUC for detecting decoding parameter mismatches (temperature, top-$p$) across models and datasets. Each cell compares outputs under a specific decoding configuration against the default $(0.5, 1.0)$; higher values indicate stronger detectability.

## C   THEORETICAL ANALYSIS OF RUT

In this section, we provide a formal analysis to characterize the statistical power of RUT and its sample efficiency under principled assumptions about language model output distributions and adversarial perturbations.

We model two basic attack types: (1) **reparameterization attack**, which preserve ranking of tokens but smoothly alter token probabilities via a small parameter shift $s \mapsto s + \epsilon$; and (2) **reordering attack**, which randomly samples a token $t$ according to the distribution, and then swaps $t$ with the highest-probability token, thereby making $t$ the new top-1 token. Real-world attacks almost always combine both, and tend to easier to detect than either alone.

Throughout the proof, we assume that RUT uses the **log-rank** score function. Based on the famous Zip's law in statistical linguistics (Piantadosi, 2014), we assume that the output token distributions at any position follows a Zeta distribution with parameter $s$, where $s \in (1, +\infty)$ reflects the heavy-tailedness of the distribution.

It can be easily shown that the FPR of RUT is $\alpha$, the significance level that we choose. Below, we focus on type II error: the probability of missing an attack when one exists.

**Lemma C.1.** *Let the rank of a token be a random variable $K$ on $\mathbb{Z}^+$ with probability mass function $p(k) = k^{-\alpha}/\zeta(\alpha)$ for $\alpha > 1$. Let $x = \log k$. The survival function $S(x) = P(\log K \geq x)$, for large $k = e^x$, has the asymptotic form*

$$S(x) = \frac{e^{(1-\alpha)x}}{\zeta(\alpha)(\alpha - 1)}(1 + o(1))$$

*Proof.* The survival function is the probability that the rank $K$ is greater than or equal to some value $m$. By definition, this is the sum of the probabilities of all ranks from $m$ to infinity.

$$S(\log m) = P(K \geq m) = \sum_{k=m}^{\infty} p(k) = \frac{1}{\zeta(\alpha)} \sum_{k=m}^{\infty} k^{-\alpha}$$

For large $m$, this sum can be approximated by its corresponding integral. The leading term of the Euler-Maclaurin formula shows that the sum and integral are asymptotically equivalent.

$$
\begin{aligned}
\sum_{k=m}^{\infty} k^{-\alpha} &= \int_{m}^{\infty} t^{-\alpha}dt + O(m^{-\alpha}) \\
&= \left[\frac{t^{-\alpha+1}}{1-\alpha}\right]_{m}^{\infty} + O(m^{-\alpha}) \\
&= 0 - \frac{m^{1-\alpha}}{1-\alpha} + O(m^{-\alpha}) \\
&= \frac{m^{1-\alpha}}{\alpha-1} + O(m^{-\alpha}) \\
&= \frac{m^{1-\alpha}}{\alpha-1}(1 + O(m^{-1})) \\
&= \frac{m^{1-\alpha}}{\alpha-1}(1 + o(1))
\end{aligned}
$$

Substituting this result back into the expression for the survival function, we obtain

$$S(\log m) = \frac{m^{1-\alpha}}{\zeta(\alpha)(\alpha - 1)}(1 + o(1))$$

The analysis is performed on the distribution of the log-rank score, $x = \log k$. We therefore let $x = \log m$, which implies $m = e^x$. Substituting this into the expression above yields the final asymptotic form for the survival function of the log-rank.

$$S(x) = \frac{(e^x)^{1-\alpha}}{\zeta(\alpha)(\alpha - 1)}(1 + o(1)) = \frac{e^{(1-\alpha)x}}{\zeta(\alpha)(\alpha - 1)}(1 + o(1))$$

For the remainder of this analysis, we denote this survival function, which serves as the CDF for our rank-based test, by $F(x, \alpha)$. $\qquad \square$

**Lemma C.2.** *For a hypothesis test using the Cramér-von Mises (CvM) statistic to distinguish a null hypothesis $H_0$ from an alternative hypothesis $H_1$ with a fixed significance level $\alpha_{err}$ and a fixed statistical power $1 - \beta_{err}$, the required sample size $n$ is assymptotically inversely proportional to the CvM distance $\omega_\infty^2$ between the two distributions, i.e.*

$$n = \Theta\left(\frac{1}{\omega_\infty^2}\right), \ as \ \omega_{+\infty}^2 \to 0,$$

*where the constant coefficient is given by (1).*

*Proof.* Let the ranks $r_i$ for $i = 1, \ldots, n$ be drawn from a distribution with CDF $F(x)$. The hypothesis test is $H_0 : F(x) = U(x)$ versus $H_1 : F(x) = F_1(x)$, where $U(x) = x$ is the CDF of the Uniform[0,1] distribution. The CvM test statistic is

$$\omega^2 = n \int_0^1 (F_n(x) - U(x))^2 dx$$

where $F_n(x)$ is the empirical CDF. The test rejects $H_0$ if the observed statistic $\omega^2$ exceeds a critical value $c_\alpha$. The power of the test is $1 - \beta_{err} = P_{H_1}(\omega^2 > c_\alpha)$. The analysis of this power requires the distribution of $\omega^2$ under $H_1$.

While the distribution of $\omega^2$ under $H_0$ is a non-normal, weighted sum of chi-squared variables, its distribution under a fixed alternative $H_1$ is asymptotically normal as $n \to \infty$. As a standard result in statistics, this follows from the central limit theorems of U-statistics, which establish that the standardized statistic converges in distribution to a standard normal variable:

$$\frac{\omega^2 - E_{H_1}[\omega^2]}{\sqrt{\text{Var}_{H_1}(\omega^2)}} \xrightarrow{d} N(0, 1)$$

The expected value $E_{H_1}[\omega^2]$ contains a non-centrality parameter that grows linearly with $n$, such that $E_{H_1}[\omega^2] = n \cdot \omega_\infty^2 + O(1)$, where the CvM distance $\omega_\infty^2$ is defined as

$$\omega_\infty^2 = \int_0^1 (F_1(x) - U(x))^2 dx$$

The standard deviation, $\sqrt{Var_{H_1}(\omega^2)}$, grows slower than the mean, and the standardized deviation $\sigma_1 = \sqrt{\text{Var}_{H_1}(\omega^2)}/n$ is asymptotically $O(1)$. We then analyze the power for large $n$:

$$1 - \beta_{err} = P\left(Z > \frac{c_\alpha - (n \cdot \omega_\infty^2 + O(1))}{\sigma_1}\right)$$

where $Z$ is a standard normal variable. For the power to be a desired constant, the argument to the probability function must also be a constant, denoted $-z_\beta$.

$$\frac{c_\alpha - n \cdot \omega_\infty^2 + O(1)}{\sigma_1} = -z_\beta$$

Solving this equation for $n$ gives

$$n \cdot \omega_\infty^2 = c_\alpha + z_\beta \sigma_1 + O(1) = \Theta(1) \qquad (1)$$

Since the right-hand side consists of terms that are constant for a fixed significance level and power, it follows that $n \cdot \omega_\infty^2$ must be constant. This implies the inverse proportionality $n \propto 1/\omega_\infty^2$. $\qquad \square$

**Theorem C.1.** *Consider a reparameterization attack that perturbs the token rank distribution's Zipfian parameter $\alpha$ by an amount $\epsilon$. For RUT to achieve a constant Type II error rate:*

- *When $\alpha \to 1^+$ (moderate heavy-tailedness), a target sample size of $n = \Theta(\epsilon^{-2}(\alpha - 1)^2)$ is sufficient.*

- *When $\alpha \to +\infty$ (extreme heavy-tailedness), a target sample size of $n = \Theta(\epsilon^{-2}\alpha^5)$ is sufficient.*

*Proof.* A reparameterization attack shifts the distribution parameter from $\alpha$ to $\alpha' = \alpha + \epsilon$. The CvM distance $\omega_\infty^2$ is proportional to $\epsilon^2$ and the integral of the squared derivative of the log-rank CDF, $F(x, \alpha)$, with respect to $\alpha$. First, we derive the asymptotic form of this derivative. Let $F(x, \alpha) = e^{(1-\alpha)x} D(\alpha)^{-1}$, where $D(\alpha) = \zeta(\alpha)(\alpha - 1)$. The partial derivative is

$$\begin{aligned}
\frac{\partial F(x, \alpha)}{\partial \alpha} &= \frac{\partial}{\partial \alpha} \left( e^{(1-\alpha)x} D(\alpha)^{-1} \right) \\
&= -x e^{(1-\alpha)x} D(\alpha)^{-1} - e^{(1-\alpha)x} D(\alpha)^{-2} D'(\alpha) \\
&= -e^{(1-\alpha)x} \left( \frac{x}{D(\alpha)} + \frac{D'(\alpha)}{D(\alpha)^2} \right)
\end{aligned}$$

where

$$D'(\alpha) = \frac{d}{d\alpha}(\zeta(\alpha)(\alpha - 1)) = \zeta'(\alpha)(\alpha - 1) + \zeta(\alpha)$$

As $\alpha \to \infty$, we have the known asymptotic behaviors $\zeta(\alpha) = 1 + o(1)$ and $\zeta'(\alpha) = o(1)$. Therefore,

$$D(\alpha) = \zeta(\alpha)(\alpha - 1) = (1 + o(1))(\alpha - 1) = (\alpha - 1)(1 + o(1))$$

$$D'(\alpha) = \zeta'(\alpha)(\alpha - 1) + \zeta(\alpha) = o(1)(\alpha - 1) + (1 + o(1)) = 1 + o(1)$$

Substituting these into the expression for the derivative gives

$$\begin{aligned}
\frac{\partial F(x, \alpha)}{\partial \alpha} &= -e^{(1-\alpha)x} \left( \frac{x}{(\alpha - 1)(1 + o(1))} + \frac{1 + o(1)}{(\alpha - 1)^2(1 + o(1))^2} \right) \\
&= -e^{(1-\alpha)x} \left( \frac{x}{\alpha - 1}(1 + o(1)) + \frac{1}{(\alpha - 1)^2}(1 + o(1)) \right) \\
&= -e^{(1-\alpha)x} \Theta \left( \alpha^{-1} x + \alpha^{-2} \right)
\end{aligned}$$

The CvM distance $\omega_\infty^2$ is proportional to

$$\epsilon^2 \int_0^\infty \left( \frac{\partial F}{\partial \alpha} \right)^2 p(x) dx$$

The probability density function is $p(x) = -\frac{\partial F(x,\alpha)}{\partial x} = \frac{e^{(1-\alpha)x}}{\zeta(\alpha)}$.

$$\omega_\infty^2 = \Theta\left(\epsilon^2 \int_0^\infty \left(-e^{(1-\alpha)x}(\alpha^{-1}x + \alpha^{-2})\right)^2 \frac{e^{(1-\alpha)x}}{\zeta(\alpha)}dx\right) \qquad (2)$$

$$= \Theta\left(\frac{\epsilon^2}{\zeta(\alpha)} \int_0^\infty e^{2(1-\alpha)x}\left(\alpha^{-2}x^2 + 2\alpha^{-3}x + \alpha^{-4}\right)e^{(1-\alpha)x}dx\right)$$

$$= \Theta\left(\frac{\epsilon^2}{\zeta(\alpha)} \int_0^\infty e^{3(1-\alpha)x}(\alpha^{-2}x^2 + O(\alpha^{-3}x) + \alpha^{-4})dx\right)$$

We now analyze this integral expression in two asymptotic regimes. As $\alpha \to 1^+$, let $k = 3(\alpha-1) \to 0^+$. The integral becomes $\int_0^\infty e^{-kx}(\alpha^{-2}x^2 + O(\alpha^{-3}x) + \alpha^{-4})dx$. We use the standard identity $\int_0^\infty e^{-kt}t^n dt = n!/k^{n+1}$.

$$\text{Integral} = \alpha^{-2}\int_0^\infty e^{-kx}x^2 dx + \alpha^{-4}\int_0^\infty e^{-kx}dx + O\left(\int_0^\infty e^{-kx}x dx\right)$$

$$= \alpha^{-2}\frac{2!}{k^3} + \alpha^{-4}\frac{0!}{k^1} + O(k^{-2})$$

$$= \frac{2\alpha^{-2}}{(3(\alpha-1))^3} + \frac{\alpha^{-4}}{3(\alpha-1)} + O((\alpha-1)^{-2}) \qquad (3)$$

As $\alpha \to 1^+$, the dominant term is $\Theta((\alpha-1)^{-3})$. Since $\zeta(\alpha)^{-1} = \Theta(\alpha-1)$, the CvM distance is

$$\omega_\infty^2 = \Theta\left(\epsilon^2 \cdot (\alpha-1) \cdot (\alpha-1)^{-3}\right) = \Theta\left(\epsilon^2(\alpha-1)^{-2}\right)$$

By Lemma C.2, the required sample size is $n = \Theta(\epsilon^{-2}(\alpha-1)^2)$.

As $\alpha \to +\infty$, a re-calculation results in (2) being of order $\Theta(\alpha^{-5})$; note that (3) no longer apply due to change in the limiting condition. Since $\zeta(\alpha) = 1 + o(1)$, we have

$$\omega_\infty^2 = \Theta\left(\epsilon^2 \cdot \alpha^{-5}\right)$$

By Lemma C.2, the sample size is $n = \Theta\left(\epsilon^{-2}\alpha^5\right)$. $\qquad\square$

**Theorem C.2.** *Consider a reordering attack where a sampled token is made the new top-1 token. For RUT to achieve a constant Type II error rate:*

- *When $\alpha \to 1^+$ (moderate heavy-tailedness), a target sample size of $n = \Theta(1)$ is sufficient.*

- *When $\alpha \to +\infty$ (extreme heavy-tailedness), a target sample size of $n = \Theta(2^\alpha)$ is sufficient.*

*Proof.* For a reordering attack, the CvM distance $\omega_\infty^2$ is the expected squared difference in the log-rank scores, given by

$$\omega_\infty^2 = \sum_{k=1}^{+\infty} \frac{k^{-\alpha}(1 - k^{-\alpha})\log k}{\zeta(\alpha)^2}$$

Expanding the numerator and splitting the expression into two sums yields

$$\omega_\infty^2 = \frac{1}{\zeta(\alpha)^2}\left(\sum_{k=1}^\infty \frac{\log k}{k^\alpha} - \sum_{k=1}^\infty \frac{\log k}{k^{2\alpha}}\right)$$

Using the identity $\zeta'(s) = -\sum_{k=1}^\infty k^{-s}\log k$, we obtain

$$\omega_\infty^2 = \frac{1}{\zeta(\alpha)^2}\left((-\zeta'(\alpha)) - (-\zeta'(2\alpha))\right) = \frac{\zeta'(2\alpha) - \zeta'(\alpha)}{\zeta(\alpha)^2}$$

We analyze this closed-form expression. As $\alpha \to 1^+$, we use the Laurent series expansions $\zeta(\alpha) = (\alpha - 1)^{-1} + \gamma + O(\alpha - 1)$ and $\zeta'(\alpha) = -(\alpha - 1)^{-2} + \gamma_1 + O(\alpha - 1)$, where $\gamma$ and $\gamma_1$ are Stieltjes constants. The term $\zeta'(2\alpha)$ converges to the finite constant $\zeta'(2)$. The numerator is dominated by $-\zeta'(\alpha)$, so

$$\omega_\infty^2 = \frac{-(\alpha - 1)^{-2} + O(1)}{((\alpha - 1)^{-1} + O(1))^2} = \frac{-(\alpha - 1)^{-2}(1 + O((\alpha - 1)^2))}{(\alpha - 1)^{-2}(1 + O(\alpha - 1))^2} = 1 + O(\alpha - 1) = \Theta(1)$$

By Lemma C.2, a sample size of $n = \Theta(1)$ is sufficient.

As $\alpha \to +\infty$, we have $\zeta(\alpha) = 1 + o(1)$. The derivative is dominated by its leading term from the series expansion, $\zeta'(\alpha) = -2^{-\alpha} \ln 2(1 + o(1))$. The term $\zeta'(2\alpha) = -4^{-\alpha} \ln 2(1 + o(1))$ is of a lower order. Therefore,

$$\begin{aligned}
\omega_\infty^2 &= \frac{-\zeta'(\alpha)(1 - \zeta'(2\alpha)/\zeta'(\alpha))}{(\zeta(\alpha))^2} \\
&= \frac{-(-2^{-\alpha} \ln 2(1 + o(1)))(1 - \frac{-4^{-\alpha} \ln 2(1+o(1))}{-2^{-\alpha} \ln 2(1+o(1))})}{(1 + o(1))^2} \\
&= \frac{2^{-\alpha} \ln 2(1 + o(1))(1 - 2^{-\alpha}(1 + o(1)))}{(1 + o(1))^2} \\
&= 2^{-\alpha} \ln 2(1 + o(1)) = \Theta(2^{-\alpha})
\end{aligned}$$

By Lemma C.2, the required sample size is $n = \Theta(2^\alpha)$. $\qquad\square$

