# OpenReview forum: "Auditing Black-Box LLM APIs with a Rank-Based Uniformity Test"
_ICLR.cc/2026/Conference — ICLR 2026 Poster_

### Official Review · Reviewer_1Ptp · 2025-10-27

**Soundness:** 3
**Presentation:** 4
**Contribution:** 3
**Rating:** 6
**Confidence:** 3

**Summary:**

This paper proposes a Rank-based Uniformity Test (RUT) for verifying whether a black-box LLM behaves identically to a locally deployed authentic model. The extensive experiments demonstrate that the proposed method is robust to quantization, harmful fine-tuning, jailbreak prompts, and even full model substitution.

**Strengths:**

This is an interesting and solid idea, and the theoretical formulation is elegant and well grounded. The method is flexible and can be applied across different scenarios (even though its performance sometimes appears less strong, which is understandable given the introduction of probabilistic substitution attacks).

**Weaknesses:**

- When I first saw the title, I actually thought of another line of work also related to auditing LLM APIs. In that problem setting, the auditor’s goal is to identify which base model or API underlies a released service, even when the service uses a fine-tuned version. The motivation there is to protect intellectual property and ensure robustness of auditing across model variants. Your task, however, seems slightly different. You aim to verify whether a claimed model has been altered, which emphasizes sensitivity rather than robustness. The distinction between these two auditing objectives initially caused some confusion. I think the two methods are not interchangeable, so it would be very helpful if you could more clearly describe their similarities and differences in the related work section. Although some fingerprinting-based studies are mentioned, the difference between the two problem settings is not clearly explained. It would also help if Section 3 could further clarify the specific auditing scenario considered in this paper.

- My second concern relates to the applicability of the proposed method. The paper shows that RUT can easily detect discrepancies when jailbreak prompts are introduced, but this raises another question. It is quite common for service providers to add system instructions to improve model performance or to embed watermarks in outputs to protect intellectual property and prevent model misuse or content leakage. These are entirely legitimate and even responsible practices, which do not involve replacing the claimed model. Would such cases also be flagged as not the claimed model? If so, that seems problematic, as the underlying motivations are reasonable. Since many real-world services adopt these practices, such high sensitivity could make the method less practical. It may end up flagging a large number of legitimate services as inconsistent, thereby limiting its real-world utility.

**Questions:**

- What are their similarities and differences between these two tasks?
- What if the service provider adds system instructions or watermarks?

---

> ### Author Response · Authors · 2025-11-24
>
> We thank the reviewer for their insightful review and suggestions. We are glad that they found our idea “interesting and solid” and that the theoretical formulation is “elegant and well-grounded”. We have responded to their individual points below.
>
> > The distinction between these two auditing objectives initially caused some confusion… What are similarities and differences between these two tasks?
>
> We agree that there are two distinct tasks:
> - Fingerprinting work typically tackles **model identification**: given an unknown service, determine which base model or API family underlies it. The methods are  ideally robust to fine-tuning or compression, which is opposite from our goal. Many such methods are **active**, in the sense that they require provider adoption (e.g., embedding special watermarks, triggers, or fingerprints into the model or service).
> - Our work instead focuses on **model equality testing**: given black-box access to a target API and a local reference model, decide whether the target behaves the same as this reference. Here we deliberately emphasize sensitivity to different types of target model deviations (quantization, SFT, routing, hidden system prompts, etc.), and our method is **passive**, because an auditor can run it unilaterally using only standard API access plus a local reference.
>
> We have updated the related work section to make this distinction explicit.
>
>
> > What if the service provider adds system instructions or watermarks?
>
> Thank you for raising this important question! In RUT, our main goal is to answer the statistical question “are the outputs from the target model consistent with the reference model at a specific level?” This does not itself decide whether any detected deviation is acceptable or unacceptable; that is a policy decision for the auditor.
>
> In deployment, there are many ways to mitigate the concern about “legitimate” modifications, for example:
> - Firstly, the reference model can include benign, documented changes, such as safety system prompts / role prompts. Since these practices are reign, it is reasonable to expect providers to disclose them when serving a model.
> - Secondly, the scenarios we primarily target are **API services**, where API callers have full control over the system prompts and decoding configuration, rather than opaque consumer **chat bot UIs**, where users might not know the system prompt. In the API setting, any undocumented change (e.g., extra system prompts) is precisely what auditors may want to detect.
>
> Thus, RUT’s sensitivity does not inherently create “false alarms”. Instead, once a suitable reference model is agreed upon, RUT simply flags deviations from that declared model, while auditors decide how strict they want to be. We will add these clarifications into our camera ready version.

---

> > ### Comment · Reviewer_1Ptp · 2025-11-28
> > **Response to Authors**
> >
> > Thanks for the clarification and for your efforts on the related work! I also agree that API services should enforce stricter transparency regarding any undocumented changes, including additional system instructions. One thing I’m curious about is whether the proposed method can still be applied when the audit target is a chatbot or an LLM-powered application. Could we extend the tool, for example, by adjusting its tolerance for deviations?

---

> > > ### Author Response · Authors · 2025-12-03
> > >
> > > We thank the reviewer for their prompt follow-up and thoughtful question.
> > >
> > > > whether the proposed method can be applied when the audit target is a chatbot or an LLM-powered-application
> > >
> > > This is an interesting setting, but one that remains to be explored. Some LLM-powered applications (e.g. Copilot, Cursor, Perplexity.ai) allow users to select open-source models, in which case RUT *may* be applicable in principle. However, these systems typically introduce additional layers, such as agentic pipelines and retrieval modules, that  are not exposed to the user, and these components can substantially transform the raw model’s behavior.
> > >
> > > As a result, the auditing target becomes the entire application stack, not just the underlying model, and it is unclear how much deviation should be treated as expected versus anomalous. We think extending RUT with controlled tolerance is an open and interesting direction for future work.

---

### Official Review · Reviewer_wtoi · 2025-10-28

**Soundness:** 3
**Presentation:** 3
**Contribution:** 4
**Rating:** 8
**Confidence:** 3

**Summary:**

The paper devises an improved statistical test procedure for detecting if textual model responses served to a user vary distributional in a significant way from a reference model, such that one should assume the model and its system has been replaced or modified (including by variations to a system prompt).

**Strengths:**

+ The idea is fairly nifty, and appears to be a meaningful (if minor) improvement on other pre-existing techniques listed.
+ Discrimination works with relatively few samples and limited assumptions (though there are some weaknesses in those)
+ The evaluation is mostly fair and appropriate
+ There is some potentially for this to be practically useful in comparing model APIs across providers under certain assumptions

**Weaknesses:**

- Could improve the evaluation of the false-alarm scenario, where a model is compared to itself (currently only showing self-pairs)
- Worth showing the variance in evaluation with multiple draws of the 100 samples
- Some minor issues with metrics and choices in the eval

**Questions:**

I'm largely fairly happy with this paper, within the bounds it sets itself.

There could be a little more sophistication in the evaluation, but this is far enough outside my expertise that I struggle to provide concrete suggestions myself—so take the below with a grain of salt.

My sense is that there's a minor conflict in choosing the scoring function on wildchat and then evaluating it against wildchat without a hold-out set. This is somewhat remediated by the test on other datasets. However, this only impacts the robustness of the choice of f and not the overall technique.

The other evaluation critique I have is with the choice of AUC as the key metric used for the final evaluations. A stealthy "attacker" might choose to route a relatively small number of prompts, and evaluating on AUC down-weights this regime. Could consider: power at small q values. Partial AUC integrated over small q. The smallest q where distinguishing is effective.

Finally, worth being explicit about the threat model this paper is describing: you have to have access to the reference model, which narrows the scenarios under which this is useful. Likewise, with system prompt substitutions, this presumes there really is a true reference.

Despite these critiques, I'm positive about the paper and thought it was appropriately scoped and evaluated. Room for improvement, but overall I think this result is one that should be in the literature. Good luck!

---

> ### Author Response · Authors · 2025-11-24
> **Official Comment by Authors - Part 1**
>
> We thank the reviewer for their helpful review and suggestions. We are glad that they found our idea meaningful and the proposed method “practically useful”. We have responded to their individual points below.
>
> > Worth showing the variance in evaluation with multiple draws of the 100 samples
>
> We thank the reviewer for this important suggestion. We have added statistical significance reporting in the paper. Specifically, for all experiments in Sections 5.2 (quantization) and 5.3 (jailbreaking), we now report:
> - **95% confidence intervals for statistical power AUC**, computed via bootstrapping. You can see the results on updated table 2.
> - **95% Wilson confidence interval  for statistical power at each fraction $q$**, now included in plots in Appendix B.1. and Appendix B.2.
>
> For convenience, we also reproduce the updated AUC results below. Confidence intervals are shown as $(-\text{lower}, +\text{upper})$, scaled by $\times 10^{-4}$.
>
> ### Detecting Quantization
>
> | Model        | RUT            | MMD            | KS             |
> |--------------|----------------|----------------|----------------|
> | Gemma–4bit   | 0.392 (-84, +103) | 0.214 (-110, +112) | 0.017 (-32, +34) |
> | Gemma–8bit   | 0.049 (-59, +59)  | 0.043 (-62, +64)  | 0.001 (-08, +10) |
> | Llama–4bit   | 0.642 (-84, +82)  | 0.625 (-89, +93)  | 0.474 (-88, +89) |
> | Llama–8bit   | 0.132 (-82, +81)  | 0.158 (-110, +109) | 0.005 (-17, +17) |
> | Mistral–4bit | 0.586 (-95, +94)  | 0.500 (-92, +97)  | 0.330 (-100, +102) |
> | Mistral–8bit | 0.049 (-56, +61)  | 0.090 (-77, +84)  | 0.006 (-20, +22) |
>
> ----
> ### Detecting Jailbreaking
>
> | Model   | Prompt   | RUT              | MMD              | KS               |
> |---------|----------|------------------|------------------|------------------|
> | Mistral | Dan      | 0.895 (-46, +45) | 0.802 (-63, +55) | 0.873 (-37, +36) |
> | Mistral | Anti-Dan | 0.893 (-00, +00) | 0.781 (-00, +00) | 0.872 (-00, +00) |
> | Mistral | Evil-Bot | 0.892 (-44, +44) | 0.766 (-56, +59) | 0.873 (-37, +37) |
> | Gemma   | Dan      | 0.888 (-43, +44) | 0.757 (-65, +68) | 0.867 (-34, +36) |
> | Gemma   | Anti-Dan | 0.858 (-49, +53) | 0.816 (-60, +52) | 0.854 (-31, +31) |
> | Gemma   | Evil-Bot | 0.893 (-43, +42) | 0.753 (-65, +63) | 0.871 (-36, +36) |
> ----
>
> We will update all  confidence intervals in the camera-ready version.

---

> > ### Author Response · Authors · 2025-11-24
> > **Official Comment by Authors - Part 2**
> >
> > > There is minor conflict in choosing the scoring function on wildchat and then evaluating it against wildchat without a hold-out set. This is somewhat remediated by the test on other datasets.
> >
> > We thank the reviewer for pointing this out. We agree that selecting the score function on WildChat and then evaluating on the same distribution may introduce a mild risk of overfitting. As noted, our additional evaluations on BigCodeBench and MATH (Section 5.6) help mitigate this concern by demonstrating that the chosen log-rank score generalizes well across domains.
> >
> > We also include a theoretical analysis of RUT’s statistical behavior in Appendix C when using the log-rank score function. We model two representative attack paradigm: *reparameterization attack* and *reordering attack*, and derive explicit error bounds and sample complexity. These results show how the audit scales with attack perturbation level and with the heavy-tailed structure of the target language domain, which do not rely on any properties unique to WildChat.
> >
> > > A stealthy "attacker" might choose to route a relatively small number of prompts, and evaluating on AUC down-weights this regime. Could consider: power at small q values. Partial AUC integrated over small q. The smallest q where distinguishing is effective.
> >
> > We appreciate this suggestion. While we agree that performance at low substitution rates $q$ is important for catching highly stealthy attacks, performance at higher $q$ is also operationally meaningful—for example, when an API provider substitutes a cheaper quantized model to save compute, the benefit gained is proportional to the substitution rate. Thus, both low- and high-q performance matter in practice.
> >
> > To directly assess detection sensitivity at low $q$, we extracted from the statistical-power curves in Appendix B the minimum substitution rate $q$  at which each method achieves statistical power $> 0.5$ across jailbreaking settings and early-epoch SFT checkpoints.
> >
> > | Setting | RUT ($q_{min}$) | MMD ($q_{min}$) | KS ($q_{min}$) |
> > |--------|------------------|------------------|------------------|
> > | Gemma–Dan | **0.2** | 0.3 | 0.2 |
> > | Gemma–Anti-Dan | **0.2** | 0.2 | 0.2 |
> > | Gemma–Evil-Bot | **0.2** | 0.4 | 0.2 |
> > | Mistral–Dan | **0.2** | 0.3 | 0.2 |
> > | Mistral–Anti-Dan | **0.2** | 0.4 | 0.2 |
> > | Mistral–Evil-Bot | **0.2** | 0.3 | 0.2 |
> > | Llama–Alpaca SFT (ckpt 1) | **0.4** | 0.6 | 0.7 |
> > | Llama–BeaverTails SFT (ckpt 1) | **0.4** | 0.5 | 0.5 |
> > | Mistral–Alpaca SFT (ckpt 1) | **0.2** | 0.4 | 0.2 |
> > | Mistral–BeaverTails SFT (ckpt 1) | **0.2** | 0.4 | 0.2 |
> >
> > Across all settings, **RUT reaches effective detection at the lowest or tied-lowest substitution rate**, indicating that it is the  most sensitive method  when only a small portion of traffic is substituted. We will incorporate this clarification and can additionally report selection method performance at low $q$ in the camera-ready version if helpful.
> >
> > >   worth being explicit about the threat model this paper is describing: you have to have access to the reference model, which narrows the scenarios under which this is useful
> >
> > Thank you for raising this important point. Our threat model is indeed centered on settings where the auditor has access to a reference implementation of the target model including its logits. This reflects our primary motivation: auditing API host that claim to serve a specific open-weight model but may substitute or modify it in undisclosed ways. In this context, access to the reference model is both natural and standard.
> >
> > We agree that auditing first-party, closed-source APIs (e.g., ChatGPT-like systems) presents a different and equally important challenge. While RUT could in principle be applied using trusted cached outputs, these APIs involve additional complications—such as opaque system prompts, dynamic updates, and lack of a verifiable reference—that require separate methodological considerations. We therefore view auditing closed-source APIs as a complementary direction for future work.
> >
> > We have added a *limitation and future work* paragraph in the conclusion section to clarify the intended scope of our threat model.

---

> > > ### Comment · Reviewer_wtoi · 2025-11-26
> > >
> > > Thanks for incorporating some of that analysis, and text—I think it helps clarify the paper a fair bit.
> > >
> > > There's also always a tricky tension between selling work to get accepted and maximising clarity with limitations, and I tend to prefer the latter. I suspect other reviewers also prefer it that way! The changes to clarify the limitations directly in the paper are good.
> > >
> > > However, I think some of my suggestion was more targeted at the introductory framing: just make it crystal clear at the start what the attacker has access to, such that a reader really can't miss it. Then add perhaps a sentence or two about where this is realistic and where it's not.
> > >
> > > I at least would always favor papers that do something more limited but meaningful within their scope, than papers that are less upfront about applicability. Eg; an attack or defence mechanism may push the state of the art a bit, even where not wholly successful in the most relevant scenarios.
> > >
> > > Overall, I'm happy with where the paper sits, though I do note one of the other reviewer's concerns around novelty. I recognise your reply to that comment as well (that the stats isn't the point), but I think on balance the reviewer's note does reduce for me a little the size of the "contribution" piece. Nonetheless, nice paper!

---

> > > > ### Author Response · Authors · 2025-12-03
> > > >
> > > > Thank you for your continuing support of our work and your thoughtful comments!
> > > >
> > > > > making it clear the access the attacker has at the introductory framing
> > > >
> > > > We thank the reviewer for raising this helpful suggestion on writing and presentation.
> > > >
> > > > To clarify, our intended threat model already assumes access to a locally deployed reference model, and the assumption is explicitly stated in the abstract, introduction, and the problem formulation sections. However, we recognize that because these statements are distributed across different parts of the paper, a reader skimming the introduction may still miss the centrality of this requirement. In response, we have revised the introduction to include a standalone paragraph that foregrounds this assumption clearly. You can find this clarification in the latest paper revision.

---

> > ### Comment · Reviewer_wtoi · 2025-11-26
> >
> > Thanks authors, these figures look reasonable (and thank you for also including them here for clarity).
> >
> > As I already hypothesised the outcome would be reasonable, and this was just an oversight in reporting, this won't result in a change of score, but still a happier reviewer! Paper was already worthwhile in my book.

---

### Official Review · Reviewer_6Aub · 2025-10-30

**Soundness:** 1
**Presentation:** 2
**Contribution:** 1
**Rating:** 2
**Confidence:** 5

**Summary:**

This paper introduces the Rank-based Uniformity Test (RUT), a statistical method for auditing whether a deployed LLM API matches a claimed reference model. The key idea is to evaluate the rank percentile of an API’s response under the reference model’s response distribution and test for uniformity via the Cramér–von Mises statistic. The authors claim RUT is query-efficient, robust to adversarial rerouting, and effective under various threat models (quantization, jailbreak prompts, fine-tuning, and full model replacement). Extensive experiments are reported across open-weight models (Llama, Gemma, Mistral) and simulated API providers, showing RUT’s superior detection power over baselines such as Maximum Mean Discrepancy (MMD) and Kolmogorov–Smirnov (KS) tests.

**Strengths:**

1. The authors proposed a conceptually simple yet general test based on rank uniformity, adapting classical nonparametric tests (CvM, rank statistics) to LLM auditing.

2. Experiments are extensive, covering several realistic scenarios (quantization, jailbreak, SFT).

3. Implementation details are clear, with ablations on score functions and comparisons to reasonable baselines (MMD, KS).

4. Figures and tables (especially Table 2, Figure 4) effectively summarize empirical trends.

**Weaknesses:**

1. The proposed RUT is essentially an application of probability integral transform + CvM uniformity testing on log-rank scores — standard tools in nonparametric statistics. The methodological leap from MMD or KS is small; the main difference lies in choice of feature (log-rank) rather than test design. The paper frames RUT as “novel,” but it’s largely a recombination of well-known components. There is little theoretical innovation beyond empirical tuning.

2. While the paper claims higher “statistical power,” the advantage appears numerically small and inconsistent across settings (see Table 2a, where 8-bit quantization detection remains near random). The results lack statistical significance testing (e.g., confidence intervals for AUC). Moreover, RUT’s improvements might stem from using richer local sampling (100× reference draws per prompt) rather than an inherently stronger test.

3. The approach requires many reference samples per prompt (m=100), which is computationally expensive and unrealistic for auditing large APIs. The “query efficiency” claim is therefore misleading—it’s query-efficient only w.r.t. API calls, not total inference cost.
The assumption that the same decoding parameters and tokenizer are available is very strong; real APIs often obscure these details. The paper briefly mentions this but does not evaluate robustness under parameter mismatch or tokenization drift.
The adversarial rerouting model is simplistic (Equation 3) and not validated against actual API behaviors.

4. No analytical characterization is provided for Type-I/Type-II error bounds or sample complexity under common substitution settings. The “uniformity under H₀” claim is intuitive but lacks formal proof of robustness when Fπref is empirically estimated. Without finite-sample guarantees, the method’s reliability remains unclear.

5. Recent works such as Cai et al. 2025 (Are You Getting What You Pay For?) and Gu et al. 2025 (Auditing Prompt Caching) are only cited but not empirically compared. Many state-of-the-art auditing or watermarking methods (e.g., fingerprint-based ones like Pasquini et al. 2024) are ignored in experiments, weakening the completeness of the evaluation.

6. RUT provides only a binary “reject or not” decision with no explanation of why a model differs. In practical auditing, it is crucial to pinpoint whether deviation stems from quantization, fine-tuning, or system prompt injection. The method offers no such diagnostic insight, limiting its operational usefulness.

7. The paper repeatedly claims “robustness to adversarial rerouting” but provides no adversarial evaluation where the provider actively detects audit traffic. All experiments are offline simulations assuming passive substitution. The claim is therefore unsubstantiated.

8. Several figures (especially AUROC plots in Appendix A.2) provide redundant visualizations without interpretation. The statistical terminology (e.g., “rank percentile under empirical CDF”) is used loosely. The method’s dependence on randomization (U~Uniform[0,1]) introduces variance but isn’t analyzed.

**Questions:**

1. How sensitive is RUT to tokenization mismatches or decoding parameter drift between π_ref and π_tgt? Could small discrepancies trigger false positives?

2. What is the theoretical sample complexity (number of prompts or reference draws) required to detect a given substitution level with 95% confidence?

3. How does RUT perform when API responses are post-processed (e.g., truncation, moderation filtering)?

4. Can the method scale to GPT-4/Claude-level APIs where reference sampling (m=100) is impractical?

5. Can you clarify whether the rank-based test is truly undetectable by adversarial rerouting? Have you simulated a provider that dynamically switches models upon detecting repeated prompts?

---

> ### Author Response · Authors · 2025-11-24
> **Official Comment by Authors - Part 1**
>
> We thank the reviewer for their insightful review and suggestions. We are encouraged that they found the proposed method is "conceptually simple yet general”. We address each of the reviewer’s concerns in detail below.
>
> >  The proposed RUT is essentially an application of standard tools in nonparametric statistics. There is little theoretical innovation beyond empirical tuning.
>
> We agree that RUT builds on classical statistical tools, and we would like  to clarify  that our  contribution is not a newer or  stronger statistical test, but an effective and practically robust auditing method tailored to the adversarial LLM API setting,  which is currently underserved in the literature.
>
> A  key innovation is that RUT requires only one API call per natural  prompt, it avoids repeated or structured queries that an adversarial provider can easily detect and reroute. Under this design, we show that RUT consistently outperforms existing baselines with the same API query budgets. Thus, although grounded in standard statistical primitives, we believe RUT constitutes a meaningful methodological advance through its demonstrated robustness, API query efficiency, and practical effectiveness.
>
> >  Limited improvement on statistical power, especially in quantization setting in section 5.1
>
> We would like to argue that RUT consistently achieves  higher power AUC across adversarial scenarios  including jailbreaking prompts, SFT models, and full model replacement. And in the more challenging quantization scenarios  of Sections 5.2 and 5.6, RUT outperforms the baselines  in 9 out of 12 settings  where the methods get meaningful detection.
>
> We appreciate the reviewer’s observation on the quantization case, and would like to provide an additional  study: we find that 8-bit quantized Llama model  induces more surface-text level drifts (Hamming KS test rejects \~59% prompts) than log-rank drift (\~41%). Since RUT is based on log-rank drift while MMD is based on text-level drift, this explains RUT’s underperformance in the 8-bit case.
>
> Specifically, we compared the original Llama outputs $Y_{ref}$ with its 8-bit quantized version outputs $Y_{8bit}$. For each prompt, we perform:
> Logrank KS test: compare logrank distributions of $Y_{ref}$ and $Y_{8bit}$.
> Hamming KS test: compare hamming similarity distributions from $(Y_{ref}, Y_{ref})$ and $(Y_{ref}, Y_{8bit})$.
>
> Across prompts, we obtained two sets of KS p-values:
> - Logrank KS: **mean p = 0.289, median p = 0.1548, proportion p < 0.05 = 41%**.
> - Hamming KS: **mean p = 0.198, median p = 0.0167, proportion p < 0.05 = 59%**.
>
> The higher detection rate in the hamming test suggests that 8-bit quantization introduces more surface-form text differences than log-rank shifts, likely explaining MMD's stronger performance in this setting. RUT’s underperformance in this setting can be potentially mitigated through increasing sample size or using a composite score function combining textual and logit-level differences, enabling it to capture a broader range of shifts.

---

> ### Author Response · Authors · 2025-11-24
> **Official Comment by Authors - Part 2**
>
> >  The results lack statistical significance testing
> > The method’s dependence on randomization (U~Uniform[0,1]) introduces variance but isn’t analyzed.
>
> We thank the reviewer for this important suggestion. We have added statistical significance reporting in the paper. Specifically, for all experiments in Sections 5.2 (quantization) and 5.3 (jailbreaking), we now report:
> - **95% confidence intervals for statistical power AUC**, computed via bootstrapping. You can see the results on updated table 2.
> - **95% Wilson confidence interval  for statistical power at each fraction $q$**, now included in plots in Appendix B.1. and Appendix B.2.
>
> For convenience, we also reproduce the updated AUC results below. Confidence intervals are shown as $(-\text{lower}, +\text{upper})$, scaled by $\times 10^{-4}$.
>
> ### Detecting Quantization
>
> | Model        | RUT            | MMD            | KS             |
> |--------------|----------------|----------------|----------------|
> | Gemma–4bit   | 0.392 (-84, +103) | 0.214 (-110, +112) | 0.017 (-32, +34) |
> | Gemma–8bit   | 0.049 (-59, +59)  | 0.043 (-62, +64)  | 0.001 (-08, +10) |
> | Llama–4bit   | 0.642 (-84, +82)  | 0.625 (-89, +93)  | 0.474 (-88, +89) |
> | Llama–8bit   | 0.132 (-82, +81)  | 0.158 (-110, +109) | 0.005 (-17, +17) |
> | Mistral–4bit | 0.586 (-95, +94)  | 0.500 (-92, +97)  | 0.330 (-100, +102) |
> | Mistral–8bit | 0.049 (-56, +61)  | 0.090 (-77, +84)  | 0.006 (-20, +22) |
>
> ----
> ### Detecting Jailbreaking
>
> | Model   | Prompt   | RUT              | MMD              | KS               |
> |---------|----------|------------------|------------------|------------------|
> | Mistral | Dan      | 0.895 (-46, +45) | 0.802 (-63, +55) | 0.873 (-37, +36) |
> | Mistral | Anti-Dan | 0.893 (-00, +00) | 0.781 (-00, +00) | 0.872 (-00, +00) |
> | Mistral | Evil-Bot | 0.892 (-44, +44) | 0.766 (-56, +59) | 0.873 (-37, +37) |
> | Gemma   | Dan      | 0.888 (-43, +44) | 0.757 (-65, +68) | 0.867 (-34, +36) |
> | Gemma   | Anti-Dan | 0.858 (-49, +53) | 0.816 (-60, +52) | 0.854 (-31, +31) |
> | Gemma   | Evil-Bot | 0.893 (-43, +42) | 0.753 (-65, +63) | 0.871 (-36, +36) |
> ----
> We will update all  confidence intervals in the camera-ready version.
>
>
>
> > RUT’s improvements might stem from using richer local sampling (100× reference draws per prompt) rather than an inherently stronger test
>
> We agree that RUT leverages more *local* sampling from the reference model, and we view this as an intentional and practical design choice rather than an inherent limitation. Our goal is to build a practically effective auditing tool tailored to the adversarial LLM API setting, in such a setting, the strongest constraint is on **target API queries**, since structured queries or repeated prompts  can be easily detected and rerouted by a dishonest provider. RUT deliberately uses **only one API  sample per prompt**, ensuring an indistinguishable and natural query distribution, while shifting the remaining workload to local sampling, where computation is inexpensive, controllable, and non-observable to the provider.
>
> This richer local sampling is what enables RUT to maintain high sensitivity, especially in **low mixing rate $q$**, where detecting subtle substitutions is intrinsically challenging. Local sampling improves the accuracy of the empirical reference distribution, which in turn strengthens the rank-based test without increasing the target-API footprint.
>
> Finally, local computation is fully amortizable: reference samples can be cached or shared across users and sessions. This makes the additional local sampling cost much more acceptable  in practice.
> > How does RUT perform when API responses are post-processed (e.g., truncation, moderation filtering)?
> Thank you for raising this point.  Post-processing of the types mentioned does not meaningfully affect RUT’s detectability.
>
> For moderation filtering, If the API blocks a response, that query is simply excluded from the audit, as no target-model output is observed. This does not bias the rank statistics and does not weaken the test.
>
> For truncation, in all experiments we explicitly cap generations at 30 tokens. An adversarial provider cannot reliably force ultra-short outputs without severely harming usability, making such truncation-based evasion impractical. Based on our experiment results, RUT shows good detection power under the consistent length control.

---

> ### Author Response · Authors · 2025-11-24
> **Official Comment by Authors - Part 3**
>
> > The approach requires many reference samples per prompt (m=100), which is computationally expensive and unrealistic for auditing large APIs. The “query efficiency” claim is therefore misleading—it’s query-efficient only w.r.t. API calls, not total inference cost.
> > Can the method scale to GPT-4/Claude-level APIs where reference sampling (m=100) is impractical?
>
> We appreciate the reviewer’s point and agree that our notion of *query efficiency* refers specifically to **target API calls**,  which are the limiting factor in adversarial auditing. We will clarify this wording in the camera-ready version.
>
> We would also like to argue that the **local inference cost is inexpensive in  practice**. In all experiments, we use `max_len=512` for prompts and `max_len=30` for generations. Using GPT-5.1 API pricing as an upper-bound estimate, the cost of generating reference samples is:
> ```
> 100 prompts * 100 samples* (512 tokens * 1.25$/1M + 30 tokens * 10$/1M) = 9.4$
> ```
> The cost of target API calls is:
> ```
> 100 prompts * 1  samples* (512 tokens * 1.25$/1M + 30 tokens * 10$/1M) = 0.094$
> ```
> which is x100 smaller than local inferences.
>
> We consider the ~$9.4 cost reasonable for a full audit reasonable, and note that the true cost is typically much lower because (1) local open-weight model calls are usually cheaper than GPT-class API, (2) reference completions can be cached and reused across audits or users.
>
> >  The assumption that the same decoding parameters and tokenizer are available is very strong; real APIs often obscure these details. The paper briefly mentions this but does not evaluate robustness under parameter mismatch or tokenization drift.
> > How sensitive is RUT to tokenization mismatches or decoding parameter drift between π_ref and π_tgt? Could small discrepancies trigger false positives?
>
> We thank the reviewer for raising this concern. We clarify and expand on both decoding parameters and tokenization below.
>
> ### Decoding parameters
>
> Many major providers (e.g., Anyscale, Together.ai, HuggingFace Inference) allow the caller to explicitly specify decoding parameters such as temperature and top-p, and users reasonably expect these settings to be faithfully respected. In this context, a mismatch in decoding behavior is itself a deviation from the declared configuration and therefore something an audit *should* detect. To evaluate robustness under such mismatch, we have a dedicated case study in Appendix B.6, where we vary temperature and top-p. Across these experiments, **RUT remains consistently more stable than MMD and KS**, demonstrating that the method does not spuriously trigger under small decoding changes and can reliably identify meaningful deviations.
>
> ### Tokenization difference
>
> Regarding tokenization, the mainstream tokenization methods BPE and Unigram are deterministic – each text string is mapped to a unique token sequence. There are stochastic tokenization methods, such as BPE-dropout, but they are primarily used during training to improve model subword understanding and are not used during inference in production APIs, especially when cross-platform deterministic performance is of more interest for the users nowadays [1]. Therefore, we think “tokenization drift” during inference time is uncommon, and we do not view the assumption of consistent tokenization as a strong one in practice.
>
> [1] “Defeating Nondeterminism in LLM Inference”, Thinking Machines, 2025.

---

> > ### Author Response · Authors · 2025-11-24
> > **Official Comment by Authors - Part 4**
> >
> > >  The adversarial rerouting model is simplistic (Equation 3) and not validated against actual API behaviors.
> >
> > We believe the probabilistic rerouting model in Equation 3  is both realistic and appropriate for the black-box API setting. Because RUT uses only naturalistic, one-shot queries, an adversarial provider **can hardly have  reliable way to detect when auditing occurs**. Under this constraint, probabilistic mixing between the claimed model and an alternative model is a practical  and effective evasion strategy available. The tunable mixing rate $q$  is also preferable, as it shows smooth changes in power curves, enabling better measurement of the selection methods on partial substitutions.
> >
> > > No analytical characterization is provided for Type-I/Type-II error bounds or sample complexity under common substitution settings.
> > >  What is the theoretical sample complexity (number of prompts or reference draws) required to detect a given substitution level with 95% confidence?
> >
> > We thank the reviewer for raising these points. In the revised draft,  We provide a formal analysis of RUT’s statistical behavior under principled assumptions about language-model output distributions in Appendix C.
> >
> > In summary, we show that:
> >
> > - **Type-I error** is exactly controlled at the chosen significance level $\alpha$, since the randomized rank statistic is $Uniform([0,1])$ when the target and reference models match.
> > - **Type-II error and sample complexity** can be analytically characterized for two representative attack families:
> >   -  *Reparameterization attacks*:  We derive explicit sample-complexity bounds showing that the required number of target API queries scales as $n = \Theta(\epsilon^{-2}(\alpha-1)^2)$ in the realistic natural language regime $\alpha \to 1^+$.
> >   - *Reordering attacks*: We show that in the natural language regime $\alpha \to 1^+$, the CvM distance remains $\Theta(1)$, so constant-size audits already achieve nontrivial power.
> >
> > These results provide Type-II error guarantees and clarify how the audit scales with attack magnitude and heavy-tailedness of the token distribution. Please see Appendix C for the full derivations and formal theorems.
> >
> > > Recent works (Cai et al. 2025; Gu et al. 2025; Pasquini et al. 2024) are not empirically compared.
> >
> > ### [Cai et al. 2025](https://arxiv.org/abs/2504.04715)
> >
> > In the latest version (version 2 on 29 Sep 2025), the paper discusses a list of methods, below we clarify how each method relates to our setting and why certain methods are excluded from empirical comparison.
> > - Text classifier: in their analysis, the method fails to reject null hypothesis in all model substitution settings
> > -  Identity prompting: in their analysis, the method fails to distinguish quantization models
> > - Model equality testing: this is the MMD we test in our paper
> > - Benchmark-based detection: in their analysis, the method fails to distinguish quantization models
> > - Greedy decoding outputs: in their analysis, the method fails to distinguish quantization models.
> > - Log probability verification: the authors provide an empirical analysis of the potential of log prob as a signal.  We tested log likelihood, the average of log prob in the response, as a score function, and it performs worse than logrank.
> > Internal activation verification: the method requires the API provider to offer a proof to each response, which the auditor verifies using a local model. This setting is fundamentally different from ours.
> >
> > ### [Gu et al. 2025](https://arxiv.org/abs/2502.07776)
> >
> > The paper shows a prompt caching attack. The attack can undermine auditing methods that depend on repeated queries. Since RUT sends only one query per prompt, **it avoids this failure mode by design**. If a query prompt is cached due to other users, this simply corresponds to our probabilistic substitution attack, where RUT has already demonstrated strong performance.
> >
> > ### [Pasquini et al. 2024](https://arxiv.org/abs/2407.15847)
> >
> > We do not include LLMmap because it addresses a fundamentally different problem from the model-equality testing scenario considered in our work. LLMmap is designed as a supervised classification method: it infers the most likely model version from a fixed list using discriminative query strategies and inference model. In contrast, our setting requires a hypothesis-testing method that compares the target API directly against a specific reference model. Moreover, we want our method to expose discrepancies arising from prompting configurations such as system prompt and prompt frameworks, while LLMmap is explicitly designed to suppress such differences.

---

> > > ### Author Response · Authors · 2025-11-24
> > > **Official Comment by Authors - Part 5**
> > >
> > > > In practical auditing, it is crucial to pinpoint whether deviation stems from quantization, fine-tuning, or system prompt injection. The method offers no such diagnostic insight, limiting its operational usefulness.
> > >
> > > We agree that distinguishing the source of response deviations is operationally valuable. Our goal in this work, however, is more fundamental: to reliably determine whether the deployed model matches the claimed reference model under adversarial conditions. We view providing diagnostic attribution of response deviations as an important but orthogonal problem.
> > >
> > > > The paper repeatedly claims “robustness to adversarial rerouting” but provides no adversarial evaluation where the provider actively detects audit traffic.
> > > > Can you clarify whether the rank-based test is truly undetectable by adversarial rerouting? Have you simulated a provider that dynamically switches models upon detecting repeated prompts?
> > >
> > > Our claim of “robustness to adversarial rerouting” refers specifically to RUT’s query pattern–indistinguishability. RUT sends one natural query per prompt, indistinguishable from ordinary user traffic, and does not rely on repeated or structured queries that an adversarial provider can easily flag. This design gives RUT built-in robustness against active detection and rerouting.
> > >
> > > While we do not simulate an oracle-level detector that perfectly recognizes audit traffic – a setting in which any black-box test becomes impossible – we evaluate the realistic counterpart: probabilistic substitution attack. Our experiments show that RUT maintains high statistical power in most of the settings, demonstrating robustness to this  general type of rerouting strategy that adversaries can feasibly deploy.

---

### Official Review · Reviewer_v4bF · 2025-11-01

**Soundness:** 3
**Presentation:** 3
**Contribution:** 2
**Rating:** 6
**Confidence:** 3

**Summary:**

The paper introduces a testing methodology to audit black-box LLM APIs. It aims to verify if a given API corresponds to a specific model or configuration, through model version, prompt cache behavior, and decoding parameters. Since commercial LLM APIs are opaque, the authors propose a statistical testing framework that uses carefully designed query-response pairs to detect discrepancies between a target (claimed) model and the API being audited. Experiment results show that the audit can detect subtle deviations from the claimed model with high sensitivity.

**Strengths:**

1. Clear motivation and practicality: the paper addresses an important and underexplored issue, the lack of transparency in commercial LLM APIs, and it works in a practical black-box setting.
2. It formalizes the auditing task as a statistical hypothesis testing problem; also, the proposed Average AUROC algorithm is intuitive and effective for summarizing model separability without requiring access to logits or gradients.

**Weaknesses:**

1. The auditing power heavily depends on how prompts are sampled. It remains unclear how different prompt types, for example, factual benign prompts vs. adversarial prompts, affect the sensitivity of the proposed method.
2. The framework requires repeated querying of both the reference and target APIs, which is expensive and may be infeasible for large-scale or continuous audits.
3. Changes in the decoding parameters, e.g., temperature or top-p, could artificially inflate AUROC values even when the underlying models are identical. This potential confound remains underexplored.

**Questions:**

1. Why was AUROC chosen over other divergence-based measures like Jensen–Shannon or Wasserstein distances?
2. Would prompts tailored to model weaknesses, for example, factual reasoning, yield more sensitive audits than random samples?
3. How does the method distinguish between genuine model differences and sampling variability caused by small changes in temperature or top-p?
4. How small a fine-tuning or low-rank adaptation can the audit reliably detect? Is there an empirical detection threshold?
5. What’s the minimal number of queries needed to achieve a statistically significant audit result? Can the method be optimized to reduce query costs?
6. If the target API changes gradually via some silent model updates, would the audit detect incremental drift or only sharp transitions?

---

> ### Author Response · Authors · 2025-11-24
> **Official Comment by Authors - Part 1**
>
> We thank the reviewer for their insightful review and suggestions. We are glad that they found the work “clearly motivated and practical” and that the statistical formulation of the task is clear. We have responded to their individual points below.
>
> > The auditing power heavily depends on how prompts are sampled. It remains unclear how different prompt types, for example, factual benign prompts vs. adversarial prompts, affect the sensitivity of the proposed method.
> > Would prompts tailored to model weaknesses, for example, factual reasoning, yield more sensitive audits than random samples?
>
> We fully agree that the choice of prompts matters; this is unavoidable for any black-box model equality test. We’d like to emphasize that our method deliberately uses **naturalistic, user-like** traffic rather than specialized **adversarial** prompts. Moreover, our experiments have already explored the effect of different types of prompts and showed that RUT is generally applicable:
> - In our main experiments we use WildChat, a large collection of real ChatGPT interactions.
> - Section 5.6 further shows that the method generalizes across very different prompt domains such as code (BigCodeBench) and math (MATH).
> That said, RUT is **agnostic to how the prompts are chosen**: any prompt set can be used, including ones tailored to known weaknesses (e.g., factuality, safety, reasoning, etc.). We agree with the reviewer’s suggestion that using such targeted prompts is a natural way to further boost statistical power if the auditor has prior knowledge about where models may differ. We will aim to clarify this point in our final revision.
>
>
> > The framework requires repeated querying of both the reference and target APIs, which is expensive and may be infeasible for large-scale or continuous audits.
>
> Our method is designed to be **API-query efficient** on the target side.
> - In all experiments, we use **one** target API call per prompt. For RUT this is 100 prompts per trial (so 100 calls to the audited API), while the additional 100 samples per prompt are drawn from the local reference model and incur no API cost.
> - Under this budget, RUT already achieves substantially higher statistical-power AUC than MMD and KS in most scenarios.
> We agree that the dominant cost is local compute, which can be amortized and cached. In practice, an auditor can:
>    1. Pre-compute or cache local reference completions for a fixed pool of prompts
>    2. Reuse them across multiple audits of different providers
>    3. Continuously accumulate rank statistics from user traffic, so that many audits require no extra target-API calls at all
>
>
> > Changes in the decoding parameters, e.g., temperature or top-p, could artificially inflate AUROC values even when the underlying models are identical. This potential confound remains underexplored.
> > How does the method distinguish between genuine model differences and sampling variability caused by small changes in temperature or top-p?
>
> First, we’d like to clarify that AUROC is not the auditing statistic. It is used only once in Section 4.2 as a metric to choose a score function (log-rank) among several candidates. The audit itself is performed using the Cramér–von Mises statistic on the rank distribution.
>
> Regarding concerns about decoding parameters, we believe that many APIs allow the caller to explicitly set temperature / top-p (e.g., Anyscale, Together.ai, HuggingFace, etc.), and users expect the requested configuration to be actually used. In this case, **a decoding mismatch is itself a violation that an audit should detect**.
>
> We also include a case study on decoding parameter mismatch in Appendix B.6 and show that RUT consistently achieves superior performance compared to baselines.

---

> > ### Author Response · Authors · 2025-11-24
> > **Official Comment by Authors - Part 2**
> >
> > > Why was AUROC chosen over other divergence-based measures like Jensen–Shannon or Wasserstein distances?
> >
> > We’d like to clarify that, in Section 4.2, we only seek to compare candidate score functions (log-likelihood, rank, log-rank, entropy, LRR) and empirically pick the one that best separates two known models under a fixed prompt distribution. For this specific purpose, we use AUROC because it matches our use case exactly (*given scores for reference vs. alternative responses, how well this score separates the two classes*) and is scale-free and threshold-free.
> >
> > We will clarify this design choice and emphasize that AUROC is used only as a proxy for score quality, not as the final test statistic.
> >
> >
> > > How small a fine-tuning or low-rank adaptation can the audit reliably detect? Is there an empirical detection threshold?
> >
> > In Section 5.4, we fine-tune LoRA models on only 500 instructions, for up to 5 epochs, using both benign (Alpaca) and harmful (BeaverTails) datasets. Figure 3 shows that RUT’s statistical power AUC rises noticeably after **just a single epoch** and remains consistently higher than MMD and KS later.
> >
> > This indicates that RUT is sensitive to relatively light LoRA fine-tuning on a few hundred examples. A full “minimal detectable change” curve over rank, dataset size, and number of epochs would require a large grid search; due to resource constraints we focused on a realistic but already subtle scenario in our paper.
> >
> >
> > > What’s the minimal number of queries needed to achieve a statistically significant audit result? Can the method be optimized to reduce query costs?
> >
> > For the number of API calls $n$, we refer to our theoretical analysis in Appendix C of our updated draft:
> >
> > - When $\alpha \approx 1$, which is well-established for natural language [1, 2], the required sample sizes are:
> >   - **Reparameterization attack:** $n = \Theta(\epsilon^{-2}(\alpha - 1)^2)$
> >   - **Reordering attack:** $n = \Theta(1)$
> >
> > These bounds are **very small** when $\alpha \to 1^+$, confirming that **RUT is very  efficient in realistic language distributions**. Please see the full derivations in Theorems C.1. and C.2. in Appendix C.
> >
> > Empirically, we show in our experiments that **100 API queries** already yield strong performance across various attack scenarios, such as detecting jailbreaking prompt, full model substitution, and SFT.
> >
> > As for the local sample size \(m\), this can be easily distributed across users. If several interested parties coordinate, they can share the workload and run evaluations in parallel. This collaborative setting not only reduces the per-participant cost to a negligible level, but also enables a more robust audit regime. Specifically, it offers two key advantages:
> > 1. The API provider cannot anticipate which prompts will be audited, forcing consistent behavior across all queries to avoid detection.
> > 2. The local compute cost is amortized across participants and sessions, enabling efficient, one-time audits without requiring continuous GPU resources.
> >
> >
> > [1] “Large-scale analysis of Zipf ’s law in English texts”, Moreno-S´anchez et al., 2015
> >
> > [2] “Zipf ’s law for word frequencies: word forms versus lemmas in long texts”, Corral et al., 2015
> >
> >
> > > If the target API changes gradually via some silent model updates, would the audit detect incremental drift or only sharp transitions?
> >
> > Yes, the audit would detect incremental drift, and our experiments are showing evidence of this from different perspectives.
> >
> > Firstly, our theoretical formulation explicitly models probabilistic substitution with a mixing rate $q$, which we vary in experiments from 0 to 1. Figure 1 shows that RUT’s statistical power increases smoothly with the fraction of traffic routed to the alternative model, which suggests that the method can detect incremental drift.
> >
> > Moreover, a gradual “silent update” can be viewed as the model on the API drifting further away from the reference over time, which our SFT experiment closely reflects. In Section 5.4, we show that more substantial deviations from the reference are indeed easier to detect: RUT’s power AUC grows with the magnitude of the target model’s behavioral change, i.e., more training epochs.

---

### Author Response · Authors · 2025-11-24
**Summary of Updates in the Revised Draft**

We thank all reviewers for their thoughtful feedback. We have updated the draft accordingly and summarize the key revisions below:
- We revised the related work section to make the **distinction from LLM fingerprinting** clearer and more explicit.
- We added a **theoretical analysis of RUT’s statistical behavior** under two representative attacks, *reparameterization attacks* and *reordering attacks*, in Appendix C.
- We added **95% confidence intervals** for both statistical power and statistical-power AUC. The updated results for detecting quantization and jailbreaking are in Table 2, with full curves in Appendix B.1 and B.2.
- We updated the introduction and added a limitation and future work paragraph in the conclusion section to **clarify the scope of our contributions**.
- All of the new or modified text is highlighted in blue in the revised manuscript.

We hope these updates address all of the reviewers’ concerns and improve the clarity and rigor of the paper.

---

### Meta-Review · Area_Chair_Wpz8 · 2026-01-07

**Summary:**

This paper proposes a rank-based uniformity test (RUT) for auditing whether a deployed black-box LLM API behaviorally matches a claimed reference model under adversarial substitution. The core idea is to assess the uniformity of rank statistics derived from reference-model likelihoods, and the method is evaluated across a range of realistic deviation scenarios, including quantization, fine-tuning, jailbreak prompting, and full model replacement.

Overall, reviewer opinions are mixed but lean positive. Multiple reviewers find the problem well-motivated and relevant to current concerns around LLM transparency and accountability. Strengths highlighted across reviews include the clear statistical formulation of the auditing task, the conceptual simplicity of the rank-based testing approach, and the breadth of experimental evaluations across models and threat settings. The paper is generally well written, with a clear experimental setup and a systematic comparison against reasonable baselines.

The main point of contention concerns novelty and methodological depth. One reviewer argues that the proposed method largely combines well-known nonparametric statistical components, and questions whether the empirical gains over existing tests are sufficiently substantial to justify acceptance. Related concerns include the computational cost associated with reference sampling and the limited scope of the threat model, which assumes access to a trusted reference implementation. These concerns are valid and worth noting. However, they appear to reflect differences in emphasis on theoretical novelty and scope, rather than identifying a fundamental flaw in correctness or experimental soundness. Other reviewers view the contribution as a careful and practically meaningful adaptation of classical statistical tools to an important and timely auditing setting, supported by extensive empirical evidence.

Taking all reviews into account, I believe the paper meets the bar for acceptance at ICLR. While the methodological novelty is incremental, the formulation is clean, the empirical study is thorough, and the problem addressed is of clear interest to the community. The identified limitations are largely acknowledged by the authors and do not undermine the core claims within the stated scope. I therefore recommend acceptance, with the assessment that this is a weak accept paper.

**Reviewer Concerns:**

The rebuttal and revised draft addressed several substantive reviewer concerns. In particular, the authors clarified the threat model assumptions (notably the requirement of access to a trusted reference model), expanded the discussion of limitations and applicability, and added additional analyses and confidence intervals that improve the transparency and rigor of the empirical results. Concerns regarding reporting omissions and experimental clarity raised by multiple reviewers were largely resolved, and these reviewers indicated increased confidence in the soundness and presentation of the work.

Some concerns remain partially outstanding. In particular, one reviewer continues to question the degree of methodological novelty, viewing the approach as a recombination of classical nonparametric statistical tools, and raises concerns about the practical cost of reference sampling and the realism of certain adversarial assumptions. While the rebuttal provides reasonable clarification and scoping, these points ultimately reflect differences in emphasis on theoretical innovation and deployment realism rather than unresolved technical flaws. Overall, the remaining concerns do not undermine the correctness of the method within its stated scope.

**Reviewer Scores:**

Reviewers who initially expressed moderate to positive evaluations (scores in the weak accept to accept range: Reviewer v4bF, wtoi, and 1Ptp) would likely maintain their original scores or increase them slightly, given that their main concerns around clarity, reporting, and experimental completeness were directly addressed.

The reviewer who gave a low score (reject: Reviewer 6Aub) is unlikely to substantially change their score, as their primary concern regarding novelty appears to be a matter of judgment rather than a misunderstanding or missing clarification. At most, this reviewer might revise their score marginally upward, but it would likely remain below the acceptance threshold.

Overall, the score distribution would likely remain mixed, but with increased confidence from the majority of reviewers following the rebuttal and discussion.

---

### Decision · Program_Chairs · 2026-01-26

Accept (Poster)